# Language Models Need Inductive Biases to Count Inductively

**Yingshan Chang & Yonatan Bisk**
Carnegie Mellon University
{yingshac,ybisk}@cs.cmu.edu

## Abstract

Counting constitutes a core skill underlying a wide range of tasks, such as formal language recognition, multi-hop reasoning and simulating algorithms. Generalizing counting inductively is central to task success on out-of-distribution (OOD) instances where testing inputs are longer than those seen in training. While there is a large body of literature reporting poor length generalization in language models, few papers have tried to distill the "reasoning" failure to the simplest case of counting failure. We aim to provide a broader picture on whether various language model architectures can a) learn to count, and b) generalize counting inductively. This work provides extensive empirical results on architectures ranging from RNNs, Transformers, State-Space Models and RWKV. We present carefully-designed task formats, auxiliary tasks and positional embeddings to avoid limitations in generalization with OOD-position and OOD-vocabulary. We find that while traditional RNNs trivially achieve inductive counting, Transformers have to rely on positional embeddings (PEs) to count OOD. Further analyses on interpreting the learned solution reveal that different PEs encode different inductive biases that facilitate counting in different task formats. As counting is the basis for many arguments concerning the expressivity of Transformers, our finding calls for the community to reexamine the application scope of primitive functions defined in formal characterizations. Finally, modern RNNs also largely underperform traditional RNNs in generalizing counting inductively, hinting at the tradeoff modern RNNs struggle to balance between parallelized training and maintaining their recurrent nature.

## 1 Introduction

"Difficulty in generalizing to *longer* instances" is a recurring theme in the discussion of Transformer limitations, regardless of the task domain (Dziri et al., 2023; Saparov et al., 2023; Zhang et al., 2023; Del'etang et al., 2022; Liu et al., 2022; Bhattamishra et al., 2020). We find that, although the notion of length may vary across domains (e.g. sequence length, recursion depth, counter states for DSAs, stack sizes for PDAs), counting is always involved as a required component to successfully handle the task. In fact, counting might be leveraged by Transformers more often than necessary as it circumvents the need to implement recurrence. For example, Liu et al. (2022) indicates that Transformers may rely on internal representations of counts to model counter languages, as a remedy for its lack of a recurrent mechanism, but failing immediately on instances with OOD counts. Further, Zhang et al. (2023) indicates that for recursive problem-solving, specialized attention heads count the recursion depth, dependent on which depth-specific solutions are learned. Therefore, counting is crucial for Transformers to perform a variety of tasks, from formal language recognition to algorithmic reasoning. And generalizing to OOD counts is crucial for handling *longer* instances. However, it remains unclear **whether Transformers can learn to count *inductively***.

On the other hand, RASP (Weiss et al., 2021), a programming language designed to mimic Transformer computations, treats counting as a primitive function based on which more complex algorithms are built (e.g. sorting, reverse, Dyck). We question the generality of counting as a primitive building block for Transformer computation. This paper conveys an important message that counting does not come effortlessly as one might expect for a primitive function. Nontrivial requirements on positional embeddings, input formats, and the amount of training have to be satisfied in order for a Transformer to learn counting in-domain. Moreover, Transformers *do not* learn to count inductively, e.g. when

the model knows `increment(50)=51`, it still cannot output the length of a 51-symbol sequence as 51 if it has only been trained on up to 50-length sequences. Notably, in this work we do a direct comparison with both modern and classical recurrent architectures to begin elucidating the source of this modern limitation, not shared by previous approaches.

We conduct extensive experiments training Transformers to count inductively. We carefully design the input-output formats, auxiliary tasks and positional embeddings to overcome the OOD-position and OOD-vocabulary issues. However, we find negative evidence. Shallow 1L or 2L Transformers struggle to generalize inductively. Successful generalization is observed with 4L Transformers, but requiring different positional embeddings for different forms of counting. Expanding our comparison to recurrent architectures, we find that RNN and LSTM succeed at everything we have asked, whereas newer RNN architectures (e.g. State-Space Models and RWKV) have degraded performance. Our work opens up attractive challenges for augmenting Transformers with a counter-equivalent mechanism, as well as hybridizing Transformer and RNN without breaking their inherent strengths. [1]

## 2 BACKGROUND

> **The inductive counting principle**: If a word in an ordered number word list refers
> to sets with cardinality $n$, then the next word refers to sets with cardinality $n + 1$
> (Rips et al., 2006; Piantadosi et al., 2012).

### 2.1 DEFINITION OF COUNTING

We define counting as the ability to map a number word to the cardinality of a set containing a corresponding number of items. The crucial inductive step requires that, having learned the mapping of the first $n$ words to the first $n$ cardinality values, one has to infer that adding one more item results in a cardinality value corresponding to the $(n + 1)^{th}$ word in the number word list. This definition of counting is extensively studied in a branch of cognitive science concerning how children learn to count (Davidson et al., 2012; Rousselle & Vossius, 2021; Sarnecka & Carey, 2008; Spaepen et al., 2018). The cognitive science research informs that children learn to count from 1 to 5 independently in early ages, then drastically generalize to the entire natural number system by inductively inferring how the number words in one's native language map to cardinality values (Wynn, 1992; Margolis & Laurence, 2008). Further, the structure of the language (e.g. avoiding special cases or change of bases) correspond to learning to count earlier in childhood (Rousselle & Vossius, 2021).

Note how counting differs from knowing the ordered list of number words: reciting the number word list constitutes an important prerequisite of counting, but establishing the mapping of numbers from the language context to the cardinal context is the core problem of interest. Also note that the complexity of number words may vary across languages. This would only affect the difficulty of learning the number word list, without changing the requirements for establishing the mapping and performing induction. Thus, the counting task studied in this paper is language-independent, and we use arabic numerals, without loss of generality, for notational consistency. To avoid confusion between number words and cardinality values, in our writings we use arabic numerals with single quotation to denote numbers in the language context (e.g. '3'), and use arabic numerals with vertical bars (e.g. |3|) to denote numbers in the cardinal context. Numbers in the language context are treated in the same way as input/output tokens in a language model, whereas numbers in the cardinal context may only appear as internal states and its exact form may vary across individuals.

### 2.2 THE TRANSFORMER ARCHITECTURE

A Transformer takes a discrete sequence as input and outputs a discrete sequence. The input and output sequences share a vocabulary $\Sigma$. The embedding and unembedding layers project a one-hot vector with dim = $|\Sigma|$ to the Transformer's hidden_dim and back to $|\Sigma|$.

Between the embedding and unembedding layers are L layers of interleaved self-attention and MLP blocks, with LayerNorm and residual connections inserted at appropriate locations. For an intuitive understanding, self-attention layers communicate information across tokens, while MLP layers allow each token to update information across the feature dimension (i.e. hidden_dim) individually.

---

[1]Code and data are released `https://github.com/zdxdsw/inductive_counting_with_LMs`

Importantly, parameters are shared across tokens, and all input tokens perform the same operation in parallel rather than sequentially. Equation 1 mathematically defines the self-attention function.

$$\text{Attn}(Q, K, V)_t = \frac{\sum_{i=1}^{T} e^{q_t^T k_i} \odot v_i}{\sum_{i=1}^{T} e^{q_t^T k_i}} \tag{1}$$

Since all input tokens perform the same operation in parallel, the Transformer does not intrinsically distinguish tokens based on positions. Thus, positional information is needed to break this symmetry.

**Sinusoidal Positional Embedding (SinePE)**(Vaswani et al., 2017) SinePE computes positional embeddings based on sine waves, which is added to token embeddings at the input layer.

**Absolute Positional Embedding (APE)**(Devlin et al., 2019) APE assigns learnable vectors to position ids $1, ..., P$, which is added to token embeddings at the input layer.

**Rotary Positional Embedding (RoPE)**(Su et al., 2021) RoPE multiplies query and key vectors by an unlearnable rotation matrix, such that the relative rotation angle between two positions captures relative position. It requires a maximum sequence length P to be predetermined.

**Scaler Positional Embedding (SPE)**(Yao et al., 2021) SPE sides aside one dimension from the Transformer's hidden_dim and inserts positional information through a scaler value. Proposed by Yao et al. (2021) who found SPE's advantage over APE in modeling *Dyck* languages with bounded depth.

**No Positional Embedding (NoPE)** NoPE denotes the vanilla Transformer without positions. Haviv et al. (2022) suggests that the causal mask could leak positional information, by potentially allowing each token to count the number of predecessors. However, this raises the same question of whether Transformers can count. Thus, our experiments with NoPE will also inform how reliable Transformers figure out absolute positions solely from causal masks.

A model easily falls apart if it has never seen the embedding for a position beyond the training length. To tackle the OOD-position issue Kiyono et al. (2021) proposes to augment the input position_ids (PIDs) with a random shift (shifted PEs) so that you start numbering positions from a random integer between 1 and P, instead of always starting from 1. This ensures that all position embeddings will be trained. Ruoss et al. (2023) proposes a more general augmentation (randomized PEs), where the PIDs for a length=k sequence is the sorted list of k integers randomly drawn from [1, P]. We empirically find that randomized PEs perform much worse than shifted PEs. Thus we adopt shifted PEs.

## 2.3 Axes of Sequential Computation in Different Architectures

The induction step of counting can be trivially afforded by sequential modeling, where "adding one more item to the set" is operationalized as the model consuming one more input, and the increment on cardinality is operationalized as unrolling one more step along the sequential axis. Table 1 summarizes the axes for sequential computation in Transformers as well as in five representative recurrent architectures. It is important to highlight the contrast between architectures dominated by parallel computation and architectures dominated by sequential computation. Our work reveals the implication of these differences on counting. In Transformers, due to the parallel processing of attention, in order for a token to build on the computation results of its predecessors, it has to proceed to the next layer. Thus, sequential computation occurs along the axis of Transformer layers. In RNNs, sequential computation is realized through state transitions. SSMs share the concept of state transition with RNNs, but have varied implementations specific to individual models. For further information, Appendix A.1 reviews the design of recurrent architectures and Appendix F.3 discusses the trade-off between recurrence and parallelization.

| Architectures | Repetitive components that realize sequential computation |
|:---:|:---:|
| Transformer | Attention + MLP Blocks |
| RNN | Matrix Multiplication + $\int$ |
| LSTM | Matrix multiplication + $\int$ + Multiplicative gating |
| S4, Mamba | Matrix multiplication |
| Linear Attention (e.g. RWKV) | Moving avg. of history with discounted weights |

Table 1: Sequential computation enjoys the reuse of computation performed in previous steps and is realized along different axes in different architectures.

| Task Design | Examples | | Training/IND Testing | OOD Testing |
|---|---|---|---|---|
| Vanilla | Input | a a a .. .. .. .. a a a | Count('a') ∈ [1, 50] | Count('a') ∈ [51, 100] |
| | Output | 1 2 3 .. .. .. .. 48 49 50 | | |
| Vanilla + Succession | Input | a a a .. .. .. .. a a a    1 2 3 .. .. .. .. 97 98 100 | Count('a') ∈ [1, 50] | Count('a') ∈ [51, 100] |
| | Output | 1 2 3 .. .. .. .. 48 49 50    2 3 4 .. .. .. .. 98 99 100 | | |
| Helper token | Input | a a a .. .. .. .. a a a    b b b .. .. .. .. b b b | Count('a') ∈ [1, 50] Count('b') ∈ [1, 100] | Count('a') ∈ [51, 100] |
| | Output | 1 2 3 .. .. .. .. 48 49 50    1 2 3 .. .. .. .. 98 99 100 | | |
| Helper token + Shifted start | Input | 31 b b b b b .. .. .. .. .. .. b b b b b b b b | Count('a') ∈ [1, 50] Count('b') ∈ [1, 100] shiftedstart+Count('a') ∈ [1,100] shiftedstart+Count('b') ∈ [1,100] | Count('a') ∈ [51, 100] shiftedstart+Count('a') ∈ [51,100] |
| | Output | 31 32 33 34 35 36 .. .. .. .. .. .. 93 94 95 96 97 98 99 100 | | |
| | Input | 20 a a .. .. .. .. a a a    66 a a .. .. .. .. a a a | | |
| | Output | 20 21 22 .. .. .. .. 61 62 63    66 67 68 .. .. .. .. 98 99 100 | | |
| Shifted start | Input | 10 a a .. .. .. .. a a a    20 a a .. .. .. .. a a a | Count('a') ∈ [1, 50] shiftedstart+Count('a') ∈ [1,100] | Count('a') ∈ [51, 100] shiftedstart+Count('a') ∈ [51,100] |
| | Output | 10 11 12 .. .. .. .. 50 51 52    20 21 22 .. .. .. .. 61 62 63 | | |
| | Input | 45 a a .. .. .. .. a a a    66 a a .. .. .. .. a a a | | |
| | Output | 45 46 47 .. .. .. .. 69 70 71    66 67 68 .. .. .. .. 98 99 100 | | |
| Modular (mod 10) | Input | a a a a a a a a a a a a a a a a a a a a a | Count('a') ∈ [1, 50] | Count('a') ∈ [51, 100] |
| | Output | 1 2 3 4 5 6 7 8 9 10 1 2 3 4 5 6 7 8 9 10 1 | | |
| Selective | Input | $a_1$ $a_2$ $a_2$ $a_1$ $a_1$ $a_1$ $a_3$ $a_1$ $a_2$ $a_1$ $a_1$ $a_2$ $a_3$ $a_3$ $a_1$ $a_3$ $a_1$ $a_1$ $a_3$ $a_2$ $a_3$ | Count('$a_1$') ∈ [0, 10] ... Count('$a_{10}$') ∈ [0, 10] Count('$a_1$') + ... + Count('$a_{10}$') ∈ [1, 50] | Count('$a_1$') ∈ [0, 10] ... Count('$a_{10}$') ∈ [0, 10] Count('$a_1$') + ... + Count('$a_{10}$') ∈ [51, 100] |
| | Output | 1 1 2 2 3 4 1 5 3 6 7 4 2 3 8 4 9 10 5 5 6 | | |
| Selective + Modular (mod 4) | Input | $a_1$ $a_2$ $a_2$ $a_1$ $a_1$ $a_3$ $a_1$ $a_2$ $a_1$ $a_1$ $a_2$ $a_3$ $a_3$ $a_1$ $a_3$ $a_1$ $a_1$ $a_3$ $a_2$ $a_3$ | Count('$a_1$') ∈ [0, 10] ... Count('$a_4$') ∈ [0, 10] Count('$a_1$') + ... + Count('$a_4$') ∈ [1, 64] | Count('$a_1$') ∈ [0, 10] ... Count('$a_4$') ∈ [0, 10] Count('$a_1$') + ... + Count('$a_4$') ∈ [65, 128] |
| | Output | 1 1 2 2 3 4 1 1 3 2 3 4 2 3 4 4 1 2 1 1 2 | | |

Figure 1: Illustration of input-output formats. Every integer, as well as $a$, $b$, $a_1, ..., a_{10}$ are individual tokens. Row 1: Vanilla counting, where each token outputs the count of $a$'s seen from the beginning of the sequence up to itself. Row 2: Vanilla counting augmented with input-output pairs that inform the order of number tokens. Row 3-4: $b$ is the helper token, to be seen with larger counts. $a$ is the main token of interest, to bes seen with restricted counts during training and tested with OOD counts. Row 7-8: $a_1, ..., a_{10}$ are distinct tokens. Each of them should maintain its own counter.

## 3 GENERAL EXPERIMENTAL SETUP

### 3.1 DATA CREATION

When generating the data, there are two important hyperparameters at play: MAX_TRAIN_SEQLEN and MAX_OOD_SEQLEN. Note, MAX_IND_SEQLEN = MAX_TRAIN_SEQLEN. Please refer to the rightmost two columns of Figure 1 for their exact values. Since loss is computed at every token, rather than only at the last token, there is no need to include shorter training sequences. In fact, all training sequences have identical lengths equal to MAX_TRAIN_SEQLEN, in order to max out supervision on larger counts. Similarly, every testing sequence has a length equals to MAX_IND/OOD_SEQLEN.

Data creation for selective counting requires additional effort to balance the distribution. As Figure 1 shows, each input sequence in selective counting can consist up to 10 unique tokens, $a_1...a_{10}$. If we sample $a_1...a_{10}$ at random, their counts, Count('$a_1$'), ..., Count('$a_{10}$'), will heavily bias towards small values. It is desirable to balance the distribution so that each of Count('$a_1$'), ..., Count('$a_{10}$') is uniformly distirbuted over [0, 10] in training. Thus, during training data generation, we upweigh sequences where some tokens have larger counts, resulting in the distribution shown in Figure A7. The distribution of Count('$a_1$'), ..., Count('$a_{10}$') in the OOD test set is skewed towards larger values because they are longer while we restrict each unique token to appear less than ten times.

## 3.2 IMPLEMENTATION

We follow the standard GPT-2 implementation[2] and train 1, 2, 4-layer Transformers to count. Our models are restricted to be shallow because counting should not take much computation if it truly serves as the primitive building block for other complex functions. Weiss et al. (2021) suggested that computing the length of a sequence can be done within one layer. We generously increase the budget to four layers. The input-output formats are summarized in Figure 1 and detailed in Section 4. Each number word is tokenized into an individual token, corresponding to whole-number tokenization in the LLM literature. We note an alternative where numbers are tokenized into digits. We opt not to use single-digit tokenization because previous studies show that when numbers are represented by multiple tokens, early Transformer layers serve to "detokenize" , while late Transformer layers take the additional reponsibility to "re-tokenize" (Elhage et al., 2022). This suggests that whole-numbers are the preferable processing units in Transformers, while single-digit tokenization adds extra complexity of mapping token spans and numbers back and forth. This work adopts whole-number tokenization to avoid conflating the complexity of counting with that introduced by tokenization.

## 3.3 CHECKPOINTING AND EVALUATION

We checkpoint and perform IND/OOD testing *every 30K steps*. The evaluation is accuracy averaged across the sequence dimension. The total length of training is typically 312.5K or 625K steps. These stopping times are empirically chosen, by which the model has either experienced a long overfitting period with plateaued testing accuracy, or already saturated to perfect. For each model, we report performance on the *best checkpoint* over the entire course of training. We find different patterns in the training and testing curves across task variants and types of positional embeddings. While IND testing scores usually increase monotonically, OOD testing scores may bump and drop if the model overfits. Due to the space limit, we only report the maximum performance along the curves as we believe the performance *upperbound* is of more interest in this study, and leave the examination of learning dynamics to future work. There is a possible connection between the bumps observed in some of our counting tasks to the grokking (Nanda et al., 2023) phenomena. While it is impossible to rule out late grokking that would have happen after we stopped our training jobs, we already allow a long patience window within the training duration of 312.5K or 625K steps. Usually, no improvement was observed in the latter half of training. Moreover, every experiment has been repeated with five seeds, which further enlarges the search range for grokking if it could ever happen. Unless otherwise noted, we report the best performance out of five seeds. Appendix B reports the median performance out of five seeds which complement the main Transformer and RNN results in Table 2 and Table A1.

## 4 COUNTING

Training a model to count inductively requires us to provide 1) An ordered number word list covering the full set of cardinality values, 2) Examples of mapping between number words and sets of objects for small cardinalities. Crucially, the full list of ordered number words should be taught **without** exposing the model to any set of objects with an out-of-distribution cardinality. Considering this, a vanilla approach would be to add the succession sequence, i.e. '1', '2', ..., to the training data. However, this is largely ineffective, as shown in Table 2-Top. This is because the model would easily master the succession sequence by modeling the bi-gram statistics, which brings no help to counting.

A more helpful approach is to teach counting with a helper token, which is seen up to the cardinality of M. The cardinality of the main object remains to be bounded by N in training. In fact, the helper token trivializes generalization. Intuitively, this task asks: "If you have learned to count bananas up to 100, but you have never seen as many as 100 apples, can you count apples up to 100?" (Figure 1, row1). Though generalization under this setting does not require induction, we view this task as a useful sanity-check because it is undesirable to establish the counting ability tied to specific objects.

Next, we propose "shiftedstart", a modification to the input-output format, to simultaneously achieve **1)** full exposure of the vocabulary (as well as its ordering) and **2)** bounded exposure of cardinalities. Given that, we can test generalization with the OOD cardinalities. Concretely, we insert a number word (k) at the initial position to shift the beginning of the output counting sequence from 1 to k+1. We illustrate this in Figure 1, row 3. Compared to the helper token setting, the shiftedstart setting

---

[2]8 heads, 1,024 dim and 4,096 MLP-dim. LR=1e-4 with 3k steps of linear warmup. Batch size is 32.

| Task | L | NoPE IND | NoPE OOD | Sine IND | Sine OOD | APE IND | APE OOD | RoPE IND | RoPE OOD | SPE IND | SPE OOD |
|---|---|---|---|---|---|---|---|---|---|---|---|
| Vanilla | 2 | 2.0 | 0.0 | 100 | 0.0 | 100 | 0.0 | 2.0 | 0.0 | 100 | 0.0 |
| + Succession | 4 | 2.0 | 0.0 | 100 | 0.0 | 100 | 0.0 | 2.0 | 0.0 | 100 | 0.0 |
| Helper Token | 2 | 100 | 100 | 100 | 76.4 | 100 | 80.6 | 100 | 100 | 100 | 92.9 |
| | 4 | 100 | 100 | 100 | 71.7 | 100 | 69.3 | 100 | 100 | 100 | 99.8 |
| Helper Token | 1 | 100 | 4.1 | 99.7 | 0.0 | 100 | 13.6 | 100 | 7.4 | 100 | 4.1 |
| + Shifted Start | 2 | 100 | 100 | 100 | 100 | 100 | 81.34 | 100 | 78.7 | 100 | 18.3 |
| | 4 | 100 | 100 | 100 | 95.6 | 100 | 100 | 100 | 100 | 100 | 99.8 |
| Shifted Start | 1 | 100 | 16.7 | 100 | 4.3 | 100 | 50.5 | 100 | 37.7 | 100 | 9.0 |
| | 2 | 100 | 25.0 | 100 | 78.7 | 100 | 27.8 | 100 | 92.5 | 100 | 57.8 |
| | 4 | 100 | 46.1 | 100 | 48.6 | 100 | 51.9 | 100 | 98.86 | 100 | 83.8 |
| Modular (mod10) | 1 | 11.8 | 11.8 | 100 | 100 | 100 | 71.2 | 11.8 | 11.8 | 100 | 8.2 |
| | 2 | 12.0 | 11.8 | 100 | 100 | 100 | 100 | 11.8 | 11.8 | 100 | 8.2 |
| | 4 | 11.8 | 11.8 | 100 | 100 | 100 | 100 | 11.8 | 11.8 | 100 | 12.1 |
| Selective | 1 | 96.0 | 9.3 | 99.5 | 68.8 | 100 | 10.6 | 99.7 | 29.9 | 99.8 | 61.3 |
| | 2 | 99.7 | 94.1 | 99.8 | 32.6 | 100 | 13.9 | 99.7 | 49.1 | 99.7 | 86.9 |
| | 4 | 99.7 | 100 | 100 | 100 | 100 | 100 | 99.7 | 52.5 | 99.4 | 98.2 |
| Selective | 2 | 92.3 | 56.4 | 96.5 | 47.4 | 99.8 | 27.2 | 98.1 | 32.3 | 99.7 | 46.8 |
| + Modular (mod4) | 4 | 97.4 | 91.8 | 100 | 98.2 | 99.9 | 97.3 | 98.5 | 39.8 | 98.2 | 54.6 |

Table 2: **Top:** When the vocabulary corresponding to OOD-cardinality is exposed via the succession sequence, models achieve perfect accuracy on reciting the **Succession** sequence, yet perform poorly on counting. This clearly show that augmenting training data with the ordered number word list offers no assistance to counting. **Middle:** We teach the model number words covering both IND and OOD counts, without exposing the model to OOD cardinalities. This is ensured via either an auxiliary task involving a **Helper Token**, or modifying the input-output format with a **Shifted Start**. **Bottom:** Transformer counting for the **Modular** or **Selective** variants (or both). Positional embeddings are augmented with random shift by default. We denote Layers, L, and In/Out-of distribution as IND/OOD. OOD accuracies are only calculated at extrapolation positions.

imposes a greater challenge since a cardinality above N is strictly absent from training data. Moreover, shifted starts discourages a model to exploit a rigid mapping from input positions to outputs — an undesirable solution that may inflate performance. Finally, we also experiment with a "Helper Token + Shifted Start" to enrich the evidence that our task design does not permit easily-hackable solutions.

Table 2-Middle shows the counting performance of Transformers, with the training data augmented with a helper token, shifted starts, or both. A helper token indeed makes the task easier, as evidenced by near-perfect OOD accuracy of all five positional embeddings. When the order of number words is only exposed by virtue of the shifted starts, only a 4L RoPE Transformer is able to generalize. The poor results for 1L and 2L models suggest that counting in Transformers may require a non-trivial computation budget. This initial result already calls into question the validity of treating counting as a primitive operation in existing papers. Further, reasoning problems that treat counting as a primitive operation would impose larger demands in order for inductive generalizations. Otherwise, instances with larger counter states should be explicitly demonstrated during training. Extrapolation to larger counter states do not trivially emerge as a result of mastering in-domain data. A more generous read is that current results relying on counting should only be interpreted as valid for in-domain settings — not as general computational engines as papers often characterize them.

**Modular Counting** In the previous section, we found that Transformers largely failed to generalize inductively except for those equipped with RoPE. This calls for the next question: If it is too hard for Transformers to simulate "unbounded counters", can they simulate modular counters — only requiring a finite counter states? Modular counting will not run into the OOD-vocabulary issue, so remedies we apply in the last section, including the helper token or shifted starts are no longer necessary. However, modular counting introduces additional complexity for modelling periodicity. We believe modular counting should be as powerful as "unbounded counting" because, for example, a stack of mod10 counters, coordinating appropriately, would give us the entire natural number system.

Table 2 row 5 shows that only APE and SinePE generalize well on modular counting. NoPE and RoPE failed catastrophically, not even fitting the training data. The failure of RoPE is particularly interesting because one would expect its formulation to inherently inform periodicity. Section 5 provides explanation. Briefly, RoPE only modifies queries and keys, which does not help with symmetry breaking of a homogeneous input where all value vectors are identical in the first place. In this sense, RoPE behaves similarly to NoPE. Appendix C provides results showing that NoPE and RoPE must rely on an explicit beginning-of-sequence token to achieve modular counting in-domain. SPE achieves perfect in-domain accuracy but breaks immediately once extrapolation is required.

**Selective Counting**  We examine whether Transformers can selectively count predecessors satisfying a condition. In the counting context, selective counting is worth exploring because when an unbounded counter is approximated via a modular counter stack, counters above the first level will have to perform selective-modular counting. More broadly, selectivity is important because observations that carry useful information are sparse — the same consideration that motivates Mamba's proposal of selective-scan (Gu & Dao, 2023). In the general form of selective-counting, a predicate function $\mathrm{pred}$ can be learned such that each token $x$ outputs the number of predecessors $x_i$ where $\mathrm{pred}(x, x_i) = \mathrm{True}$. This corresponds to the "selector_width" primitive in RASP. Our experiments simply regard the identity indicator function as the selection condition. Learning predicate functions adds complexity along an axis orthogonal to inductive counting, which we leave for future research.

Note, we remove the requirement for generalizing to OOD counts via induction on the mapping between vocabulary and cardinality, because this is the primary subject of discussion in Section 4. In this section, we focus on the additional challenge related to selectivity. We generate the training data containing ten unique tokens such that the count of each unique token ranges from zero to ten. In the testing data, we also ensure that the counts do not exceed ten. However, our testing sequences are longer than the maximum training length. Thus, the summation of counts for all tokens, as well as the range of dependency in order to perform selection, are OOD.

Table 2 row 6 shows the results for selective counting. All PEs except for RoPE succeed given 4 layers. The observation that NoPE outperforms other PEs on selective counting is interesting. It suggests that causal masking may indeed aid in symmetry breaking. And, in fact, our results indicate that PEs might unintentionally introduce exploitable shortcuts or inductive biases unfavorable to generalization. Finally, we perform experiments on selective-modular counting. We adopt a smaller base (4 instead of 10), in order to prevent a substantial growth of sequence length, since a selective counting sequence contains 10 unique tokens interleaved together, unlike a homogeneous sequence in previous counting tasks. Results are shown in Table 2 row 6, which demonstrate a clear message: only PEs — SinePE and APE — which generalizes on both modular counting and selective counting performs well on selective-modular counting. NoPE also achieves a fairly good performance on selective-modular counting, in contrast to its poor performance on modular counting. To explain this observation, NoPE's limitation on modular counting stems from its inability to break the symmetry of a homogeneous sequence. Such a limitation no longer applies to selective-modular counting as the input becomes heterogeneous. Section 5 and Appendix D,E provide further evidence.

There are two major takeaways from our Transformer counting experiments: **1)** When counting is treated as a primitive towards more complicated reasoning, it is better to cover all possible counter states in-domain, as Transformers struggle to count inductively and rely on supervised encounter with each cardinality value. **2)** Different PE schemas exhibit strength in different forms of counting. Put concisely, RoPE succeeds at unbounded counting with shifted starts; SinePE and APE generalize at both modular and selective counting; NoPE and SPE are only competitive on selective counting. Our results motivate the integration of multiple PE schemas to take advantage of orthongonal strengths.

## 5  WHY DO EACH OF THE PE STRATEGIES BEHAVE DIFFERENTLY?

We proceed to find hidden factors that account for the observed performance differences among PE schemas. We proposes two generalizable mechanisms, for modular and selective counting, respectively. Each mechanism demands particular inductive biases. Each PE schema either supports or goes against certain inductive biases. The strengths and shortcomings of each PE schema revealed in our analysis consolidate our core argument that language models need inductive biases to count inductively. Our contributions are novel in two regards: 1) recognizing unique sets of inductive biases for modular and selective counting that are plausible for a Transformer to implement — where parallel computation dominates, and 2) studying how these inductive biases are realized by PEs. We

believe our findings will both inspire theoretical studies to quantify how much expressivity is added to Transformers by separate PE schemas, and advise downstream applications on the choice of PE based on the demand for inductive biases.

**Modular Counting** The mechanism that allows for perfect generalization to the OOD test set consists of two steps: *First Token Recognition* and *Position-based Modular Subtraction*. The first step, *First Token Recognition*, is necessary to address the additional complexity introduced by the position-shift technique (Section 2.2). The model must locate the first token in each input sequence in order to figure out the position-shift value. In the second step (Equation 2), each token attends to the first token and compute the difference in their PIDs modulo 10, which gives the desired output.

$$\text{output} = \big((\text{PID}_{\text{first\_tok}}\%10) - (\text{PID}_{\text{current\_tok}}\%10)\big)\%10 \qquad (2)$$

*First Token Recognition* demands the inductive bias for breaking symmetry. Since the input sequences in our modular counting task are homogeneous, the model must leverage PEs to distinguish among identical tokens. To test for how well each PE schema supports this inductive bias, we design a first token recognition task (dubbed *first_tok_homogeneous*) whose details are described in Appendix D. We train 1L Transformers with five PE schemas and find that only APE, SinePE and SPE are able to fit the training data and generalize to unseen sequence lengths. NoPE and RoPE's inability to recognize the first token explains their failure in modular counting, as the position-shift technique renders any rigid mapping from PID to the output useless.

*Position-based Modular Subtraction* requires the capability to cluster token representations based on their PID modulo 10. Both APE and SinePE support such constructions, as evidenced by the PCA plots of hidden states. Figure A3 plots the first two principal components of intermediate states, color coded by PID modulo 10. Tight clusters indicate that the model produce close representations for tokens whose PIDs modulo 10 have the same value. Such clustering pattern is only observed for APE and SinePE, but not for SPE models. We believe that injecting positional information only through a single dimension limits an SPE Transformer's ability to build richer features based on positions, which probably explains why SPE succeeds at first token recognition but fails at modular counting.

**Selective Counting** The generalizable mechanism suitable for the Transformer architecture again consists of two critical steps: *First Token Recognition* and *Token-based Attention*. This mechanism closely resembles the construction in Chiang & Cholak (2022) for recognizing PARITY, which crucially depends on 1) a beginning-of-sequence (BOS) symbol and 2) uniform attention over tokens that are either the BOS or identical to self. Though our task format does not include a BOS, we argue that causal masking is sufficient for first token recognition, as long as the input sequence is largely heterogeneous. This is verified through a variant of the first token recognition task (dubbed *first_tok_heterogeneous*), in which the input is a shuffled sequence containing 10 unique tokens, each occurring a random number of times. We find that a NoPE 1L causal Transformer is able to generalize well on *first_tok_heterogeneous*. Appendix E provides details about *first_tok_heterogeneous* and mechanistically describes how causal masking helps to accomplish it.

The first token, once recognized, can serve the role of BOS for subsequent layers. Following Chiang & Cholak (2022), subsequent layers should construct features that represent two quantities for each token: $1/n$ and $k/n$, where $k$ is the desired output (i.e. the count of identity tokens on or before the current token) and $n$ is the total count of tokens up to the current token. Next, LayerNorm and MLP will learn the map $(1/n, \ k/n) \to k$. The key to construct representations for $k/n$ is computing attention weights purely based on token identity, regardless of PIDs. Indeed, given the task format, positional information is not needed to solve selective counting. Therefore, one of the critical factors accounting for the different performance between PE schemas is whether the model can ignore PEs. A NoPE Transformer effortlessly achieves this, thereby already generalizing well with 2L. SPE only minimally injects positional information through a single dimension, thus not imposing much difficulty when the PEs are supposed to be ignored. In that sense, SPE performs closer to NoPE, in accordance with our results in Table 2 row 6. APE and SinePE only generalize well with 4L, due to two possible reasons: 1) It requires non-trivial effort for APE/SinePE Transformers to ignore PEs. 2) PEs are actually helpful for first token recognition, a prerequisite subtask, thus complicating the picture. We additionally experiment with Selective Counting + BOS and the results corroborate our hypothesis. Explicitly feeding BOS lowers the complexity and removes the supervision signals which might be at odds with the need for disregarding PEs. Table A7 shows that both APE and SinePE are able to emulate NoPE with 1L when BOS is included.

Nevertheless, RoPE does not benefit from BOS, indicating that its struggle may majorly come from the second subtask, *Token-based Attention*. Unlike APE, SinePE and SPE, where positions affect the input representations, RoPE directly use positions to modify queries and keys. Such modifications encode a recency bias (Su et al., 2021) — an unfavorable inductive bias in this case — which we believe is hard to be escaped by the rest of the network through learning. We hypothesize that the enforcement of recency bias leads to the difficulty of implementing pure token-based attention. One indicative piece of evidence is the variation of attention scores as the input PIDs varies, keeping the same input token ids (TIDs). Figure A6 visualizes the standard error of attention score (i.e. entries of $QK^T$) between each pair of TIDs, across all PID pairs they can take. For models trained on selective counting, attention scores in RoPE subject to the largest amount of variation influenced by PIDs, while attention scores in APE are the least sensitive to PIDs. There may be other explanations for why RoPE struggles more than other PE schemas at selective counting, which is left for future work. We hope the counting tasks proposed in this work provide a lens through which inductive biases enabled by PEs that are not otherwise encoded in self-attention can be studied in isolation. Future work may extend this work by exploring how those inductive biases carry over to broader arithmetic domains.

**Shifted Start Counting**    For shifted start counting, we are unaware of a generalizable solution, which calls into question the seemingly successful generalization of RoPE 4L in Table 2 row 4. However, the ability for RoPE to generalize is fragile. Successful generalization when MAX_TRAIN_SEQLEN = 50, MAX_OOD_SEQLEN = 100 does not imply success when the ratio between them is arbitrarily extreme. In fact, we have easily challenged RoPE 4L to failure by making MAX_OOD_SEQLEN four times larger than MAX_TRAIN_SEQLEN. Table A8 summarizes the performance of RoPE 4L with different combinations of MAX_TRAIN_SEQLEN, MAX_OOD_SEQLEN, clearly demonstrating a worsening trend as the ratio makes generalization harder. The fragility of RoPE, as well as the failure of other PE schemas indicate that the OOD-cardinality issue remains unsolved, which is the core obstacle to inductive counting in Transformers. Our work raises the importance of OOD-cardinality as a harder barrier hindering generalization on inductive counting. OOD-cardinality poses a separate difficulty from OOD-position, OOD-vocabulary, or OOD-range-of-dependency problems, and shall not be confused with these problems that the literature on length generalization (Press et al., 2021; Kiyono et al., 2021; Ruoss et al., 2023; Kazemnejad et al., 2024; Anil et al., 2022; Zhou et al., 2024) has been targeting at.

## 6    COUNTING IN OTHER LM ARCHITECTURES

As counting is a fundamentally recurrent task, it is natural to validate our conditions on recurrent architectures. Both the explicit modeling of hidden state transitions, and the sequential unrolling of computation along the input sequence dimension, naturally facilitate inductive counting. Note, there exists prior work hinting at counting in such architectures (Shi et al., 2016; Suzgun et al., 2019), but not directly evaluated in a systematic comparison. We find that traditional recurrent architectures, RNN (Elman, 1990) and LSTM (Hochreiter & Schmidhuber, 1997), achieve perfect generalization with a single layer, except that RNN slightly falls short on selective counting (Table A1). This highlights that a recurrent bias is likely key for inductive counting, which is precisely what the Transformer lacks and must therefore rely on PEs as substitute.

The recent literature has seen a resurgence of modern RNN architectures (Gu & Dao, 2023; Gu et al., 2021a; Peng et al., 2023; 2024) claiming to enjoy the best of both worlds: parallelizable training, like Transformers, and recurrent inference, like RNNs. It is important to investigate whether the recurrent formulation of these architectures affords inductive counting in the same way as traditional RNNs. To this end, we experiment with three modern RNNs — S4 (Gu et al., 2021a), Mamba (aka S6) (Gu & Dao, 2023) and RWKV-v6 (aka Finch)(Peng et al., 2024) that rival Transformers on large-scale LM benchmarks. The key observation is that modern RNNs generalize much worse than traditional RNNs on counting. We suspect the reason lies in less flexible state transitions, especially for Mamba and RWKV. The very design that enables parallel training through reformulating the model into the "convolutional mode" also limits the expressivity of state transitions. As illustrated in Table 1 and Appendix A.1, while traditional RNNs apply a nonlinearity to state transitions, modern RNNs only apply matrix multiplication or linear interpolation to history states, for the sake of easy contraction of multiple sequential updates into a single computation step. A potential limitation of this design is manifested through our counting tasks, opening up questions about what architectural elements imbue the necessary inductive biases for counting, and how these can be transferred to hybrid architectures. Appendix A provides implementation details and results for counting on other LM architectures.

## 7 RELATED WORKS

**Formal Theories on Transformer Expressivity**   Transformer expressivity can be formally analyzed from the perspectives of functional libraries (Weiss et al., 2021; Lindner et al., 2023; Zhou et al., 2023), boolean circuits (Cong et al., 1996; Yang & Chiang, 2024; Strobl et al., 2024; Merrill et al., 2022), or Automata Theory (Del'etang et al., 2022; Yao et al., 2021; Liu et al., 2022; Ebrahimi et al., 2020). We expand the discussion on this large body of literature in Appendix F. Formal studies have established proofs for Transformers' inability to count in a length-generalization regime (Hahn, 2020; Bhattamishra et al., 2020) (usually in the context of modeling counter languages). However, the proofs involve assumptions such as 1) hard attention, i.e. hardmax instead of softmax, 2) infinite sequence length, 3) pure-attention architecture, i.e. without layernorm or PEs, 4) infinite or log precision. It is unclear how certain assumptions in theoretical proof translate to real applications. Although we do not contribute new theoretical results, our work complements formal studies in important ways. First, we provide empirical evidence that echoes the theoretical proofs, under a realistic setting. Second, theories on Transformer or self-attention seldomly treat different PEs as separate cases. We argue that PEs in fact encode various inductive biases the worth detailed examination. Third, PE shift is another important realistic consideration which may affect expressivity but has been simplified away in theories. Overall, our work lays the ground where future theoretical discussions may branch out according to PE types, as well as inspiring practical design choices revolving around PEs.

**Empirically assessing Transformer expressivity**   Abundant prior works have empirically studied the capacity of Transformer-based LMs. Categorizing by scale, these works include 1) testing Transformers with hand-constructed weights (Chiang & Cholak, 2022); 2) testing Transformers trained from scratch (Del'etang et al., 2022; Abbe et al., 2023; Ebrahimi et al., 2020; Zhou et al., 2023; McLeish et al., 2024); and 3) testing pretrained LMs with finetuning (Anil et al., 2022) or prompting (Zhou et al., 2022). Categorizing by task design, prior works usually adopt synthetic tasks organized into hierarchies, with a notion of complexity informed by formal languages (Del'etang et al., 2022; Zhou et al., 2023; Hao et al., 2022; Liu et al., 2022; Kazemnejad et al., 2024; Ebrahimi et al., 2020; Ruoss et al., 2023) or boolean functions (Bhattamishra et al., 2022; Abbe et al., 2023). Our work contributes to this body of empirical evidence. Our task design additionally draws inspiration from cognitive science (Rousselle & Vossius, 2021; Sarnecka & Carey, 2008). Of particular note is that Zhou et al. (2023) also studied counting, which differs from ours by definition: the input includes a start and an end token, the output is an incremental expansion, e.g. 12 16 > 12 13 14 15 16. We believe that this can be handled by mastering the succession sequence plus a termination checking. Hence, their definition of counting involves neither numbers in the cardinality context nor induction.

We additionally review the literature on modern recurrent architectures in Appendix F.3.

## 8 CONCLUSION

Building on a growing body of work on formalizing the computation in Transformers, this work investigates counting, which is believed to be a primitive function enabling a Transformer to perform a wide range of complex tasks, such as modeling counter languages (Bhattamishra et al., 2022; Hao et al., 2022; Ebrahimi et al., 2020), simulating algorithms (Anil et al., 2022; Zhong et al., 2024; Veličković et al., 2022), and tracking the depth of reasoning chain (Saparov et al., 2023). However, there is an important distinction between counting in-domain and counting infinitely, which is understudied in the literature. While counting in-domain can be achieved with various approximations, counting infinitely imposes a significant challenge concerning induction and extrapolation. We provide extensive empirical evidence showing that 1) Counting is not a primitive function of Transformer computation as others have claimed, as it may require multiple layers to succeed at counting in-domain; 2) Different positional embeddings enable out-of-domain generalization in different forms of counting. Our findings have implications for avoiding out-of-distribution counter states in practical scenarios and the promise of integrating different positional embeddings. We also extend our investigation to recurrent architectures, including both traditional and modern models. We observe that while traditional RNNs easily generalize counting inductively, no single modern RNN generalizes on all six variants of our counting tasks, implying that inductive counting not only requires a recurrent formulation, but also demands expressive state dynamics. Thus, our investigation reveals a potential limitation where modern RNN architectures pay the cost for their lauded parallel training, motivating better solutions to combine the merits of Transformer and RNNs.

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

# A  COUNTING IN OTHER LM ARCHITECTURES

## A.1  OVERVIEW OF RNNS AND SSMS

A recurrent architecture, at each step, maintains a hidden state, updates the hidden state when consuming a new input, and produces an output based on both the current hidden state. We denote these three key operations, namely reception, transition and emission, by $i \to h$, $h \to h$, and $h \to o$, respectively. A standard (Elman) RNN implements these three steps as a matrix multiplication followed by a nonlinear activation ($\sigma$). An LSTM adopts additional parameters to compute multiplicative gates for these three steps, achieving more effective forgetting and memory.

$$h_t = \sigma(\mathbf{W_{hh}}h_{t-1} + \mathbf{W_{ih}}x_t + b_h) \qquad o_t = \sigma(\mathbf{W_{ho}}h_t + b_o) \tag{3}$$

**State Space Models (SSM)**    also incorporate reception, transition and emission operations (Eq. 4). Moreover, there is a feedthrough matrix $\mathbf{W_{io}}$ directly sending information from the input to output, bypassing the hidden state.[3]  Importantly, an SSM differs from a standard RNN in omitting the nonlinearity in state transitions. This enables a reparameterization of SSMs into a convolutional form, presented in Equation 5. The kernelization trick is essential for parallelization during training.

$$h_t = \mathbf{W_{hh}}h_{t-1} + \mathbf{W_{ih}}x_t \qquad o_t = \mathbf{W_{ho}}h_t + \mathbf{W_{io}}x_t \tag{4}$$

$$o = x * \mathbf{K} \qquad \mathbf{K} = (\mathbf{W_{ho}W_{ih}}, \mathbf{W_{ho}W_{hh}W_{ih}}, ..., \mathbf{W_{ho}W_{hh}}^k\mathbf{W_{ih}}, ...) \tag{5}$$

**Linear Attention**    Zhai et al. (2021), aims to substitute dot-product attention with sub-quadratic complexity.  Linear attention initially suffered from severe performance drop, but was recently revitalized by the RWKV model family, introducing input-dependent parameters to boost performance. At the core of RWKV is the wkv operation where $q_t^T k_i$, as in attention, is replaced by a weighted sum of history states with discounted weights. A major innovation of RWKV-v6 versus previous RWKV models lies in input-dependent derivation of the decay factor $w$.

$$wkv_t = \frac{\sum_{i=1}^t \left(e^{-(t-1-i)w+k_i} \odot v_i\right) + e^{u+k_t} \odot v_t}{\sum_{i=1}^t e^{-(t-1-i)w+k_i} + e^{u+k_t}} \tag{6}$$

The discounted moving average of history states can be rewritten in a recurrent form, thereby exhibiting a flavor of state transition.

$$\text{let } \alpha_t = \sum_{i=1}^t \left(e^{-(t-1-i)w+k_i} \odot v_i\right), \text{ then } \alpha_t = e^{-w}\alpha_{t-1} + e^{k_t} \odot v_t \tag{7}$$

## A.2  EXPERIMENTS AND RESULTS

Comparing performance between Transformers, RNNs and SSMs is important because the dominance of sequential computation in RNNs or SSMs may encode inductive biases unavailable in Transformers and favorable for inductive counting. We use the same data and experiment settings described in Section 3 for RNNs and SSMs. We select the most performant hyperparameters for each individual architecture.[4]  Results are shown in Table A1. The key takeaway is that while Transformers need careful design decisions to count, the simple RNN is able to do everything we have asked, and the newer SSM architectures have degraded performance. We believe that the inferior performance of Mamba or RWKV is due to their compromised expressivity in modeling state transitions in exchange for easy adaptation into the parallel training regime.

# B  MEDIAN PERFORMANCE OUT OF FIVE RANDOM SEEDS

We sometimes observe high variances across random seeds, suggesting the difficulty for SGD to find a generalizable solution.  Table A2 & A3 report the median performance for the same

---

[3]$\mathbf{W_{ih}}$, $\mathbf{W_{hh}}$, $\mathbf{W_{ho}}$ and $\mathbf{W_{io}}$ correspond to $\mathbf{B}$, $\mathbf{A}$, $\mathbf{C}$ and $\mathbf{D}$, respectively, in the SSM community.
[4]lr$\in \{5e{-}5, 1e{-}\{4,3,2\}\}$, w_decay$\in \{0.0, 0.01, 0.001\}$, dim$\in 2^{\{5,7,10\}}$, dropout $\in \{0.0, 0.1\}$, batch=32

| Task | L | RNN IND | RNN OOD | LSTM IND | LSTM OOD | S4 IND | S4 OOD | Mamba IND | Mamba OOD | RWKV-v6 IND | RWKV-v6 OOD |
|---|---|---|---|---|---|---|---|---|---|---|---|
| Helper Token | 1 | 100 | 100 | 100 | 100 | 100 | 100 | 100 | 15.5 | 100 | 100 |
| | 2 | 100 | 100 | 100 | 100 | 100 | 100 | 100 | 12.1 | 100 | 100 |
| | 4 | 100 | 100 | 100 | 100 | 100 | 100 | 100 | 76.5 | 100 | 100 |
| Helper Token + Shifted Start | 1 | 100 | 100 | 100 | 100 | 100 | 100 | 100 | 1.5 | 100 | 3.2 |
| | 2 | 100 | 100 | 100 | 100 | 100 | 100 | 100 | 5.1 | 100 | 0.0 |
| | 4 | 100 | 100 | 100 | 100 | 100 | 100 | 100 | 100 | 100 | 100 |
| Shifted Start | 1 | 100 | 100 | 100 | 100 | 100 | 96.3 | 100 | 7.8 | 99.9 | 29.4 |
| | 2 | 100 | 100 | 100 | 100 | 100 | 100 | 100 | 99.9 | 100 | 30.8 |
| | 4 | 100 | 100 | 100 | 100 | 100 | 100 | 100 | 97.5 | 100 | 93.8 |
| Modular (mod10) | 1 | 100 | 100 | 100 | 100 | 100 | 98.3 | 91.8 | 11.5 | 100 | 8.2 |
| | 2 | 100 | 100 | 100 | 100 | 100 | 100 | 100 | 11.1 | 100 | 11.0 |
| | 4 | 100 | 100 | 100 | 100 | 100 | 100 | 100 | 16.6 | 100 | 12.8 |
| Selective | 1 | 99.3 | 94.5 | 100 | 100 | 78.3 | 59.3 | 100 | 99.8 | 100 | 100 |
| | 2 | 99.4 | 95.1 | 100 | 100 | 91.3 | 66.7 | 100 | 96.6 | 100 | 100 |
| | 4 | 97.8 | 85.3 | 100 | 100 | 98.3 | 72.9 | 100 | 97.0 | 100 | 100 |
| Selective + Modular (mod4) | 1 | 100 | 100 | 100 | 100 | 87.7 | 27.8 | 58.4 | 28.2 | 98.5 | 83.1 |
| | 2 | 100 | 100 | 100 | 100 | 92.9 | 31.4 | 99.4 | 60.7 | 99.2 | 56.5 |
| | 4 | 100 | 100 | 100 | 100 | 99.2 | 37.8 | 100 | 59.7 | 100 | 56.8 |

Table A1: Results for recurrent architectures (in-distribution, IND, and OOD). OOD accuracies are only calculated at extrapolation positions.

experiments presented in Section 4 (Table 2 & A1). There are large discrepancies between the best and median results for Transformers with SinePE, APE, or SPE. Mamba and RWKV-v6 also exhibit such instability. As such, although certain architectural choices allows for generalizable solutions, learnability remains to be a matter of chance. We hope further study on training dynamics can find out why the likelihood of learning a generalizable solution varies across architectures.

| Task | L | NoPE IND | NoPE OOD | Sine IND | Sine OOD | APE IND | APE OOD | RoPE IND | RoPE OOD | SPE IND | SPE OOD |
|---|---|---|---|---|---|---|---|---|---|---|---|
| Helper Token | 2 | 100 | 98.5 | 100 | **66.3** | 100 | **55.3** | 100 | 100 | 100 | **14.2** |
| | 4 | 100 | 100 | 100 | **59.9** | 100 | 68.4 | 100 | 100 | 100 | **65.8** |
| Helper Token + Shifted Start | 1 | 100 | 4.1 | 99.4 | 0.0 | 100 | 13.1 | 100 | 5.3 | 100 | 4.1 |
| | 2 | 100 | 100 | 100 | 99.3 | 100 | 81.3 | 100 | 48.5 | 100 | **6.8** |
| | 4 | 100 | 99.7 | 100 | **84.3** | 100 | 100 | 100 | 100 | 100 | **84.5** |
| Shifted Start | 1 | 100 | 15.3 | 99.8 | 2.7 | 100 | 40.6 | 100 | 37.4 | 99.6 | 8.7 |
| | 2 | 100 | 20.9 | 100 | **2.0** | 100 | 19.7 | 100 | 86.6 | 100 | **43.7** |
| | 4 | 100 | 39.2 | 100 | **36.7** | 100 | **26.6** | 100 | 96.6 | 100 | **30.7** |
| Modular (mod10) | 1 | 11.8 | 11.8 | 100 | 97.9 | 100 | **56.5** | 11.8 | 11.8 | 100 | 8.3 |
| | 2 | 11.8 | 11.8 | 100 | 100 | 100 | 99.5 | 11.8 | 11.8 | 100 | 8.3 |
| | 4 | 11.8 | 11.8 | 100 | 100 | 100 | 99.9 | 11.8 | 11.8 | 100 | 10.2 |
| Selective | 1 | 95.4 | 9.2 | 95.8 | **51.0** | 100 | 10.6 | 99.7 | 26.4 | 99.7 | 57.1 |
| | 2 | 99.7 | 94.1 | 99.3 | **16.3** | 100 | 13.5 | 99.7 | 48.9 | 99.7 | 81.2 |
| | 4 | 99.7 | 100 | 100 | 99.8 | 100 | 99.9 | 99.7 | 47.6 | 99.7 | 95.4 |
| Selective + Modular (mod4) | 2 | 92.2 | 55.1 | 98.7 | 37.9 | 99.7 | 27.2 | 98.1 | 32.1 | 99.6 | 46.1 |
| | 4 | 98.0 | 91.0 | 98.8 | **70.2** | 99.3 | 95.0 | 98.5 | 38.6 | 98.2 | 49.9 |

Table A2: Results for Transformers corresponding to Table 2, switching from best to **median** performance out of 5 runs. Bold entries indicate a >10 absolute gap between the median and best results.

| Task | L | RNN | | LSTM | | S4 | | Mamba | | RWKV-v6 | |
|------|---|-----|-----|------|-----|-----|-----|-------|-----|---------|-----|
| | | IND | OOD | IND | OOD | IND | OOD | IND | OOD | IND | OOD |
| Helper Token | 1 | 100 | 100 | 100 | 100 | 100 | 100 | 100 | 15.5 | 100 | **0.0** |
| | 2 | 100 | 100 | 100 | 100 | 100 | 100 | 100 | 4.5 | 100 | **2.0** |
| | 4 | 100 | 100 | 100 | 100 | 100 | 100 | 100 | **37.0** | 100 | 100 |
| Helper Token + Shifted Start | 1 | 100 | 100 | 100 | 100 | 100 | 100 | 100 | 0.7 | 100 | 0.0 |
| | 2 | 100 | 100 | 100 | 100 | 100 | 100 | 100 | 5.0 | 100 | 0.0 |
| | 4 | 100 | 100 | 100 | 100 | 100 | 100 | 100 | 100 | 100 | **44.2** |
| Shifted Start | 1 | 100 | 100 | 100 | 100 | 100 | 95.9 | 100 | 5.2 | 99.9 | 25.0 |
| | 2 | 100 | 100 | 100 | 100 | 100 | 100 | 99.8 | **48.5** | 100 | 30.8 |
| | 4 | 100 | 100 | 100 | 100 | 100 | 100 | 100 | **51.7** | 100 | 84.0 |
| Modular (mod10) | 1 | 100 | 100 | 100 | 100 | 100 | 97.8 | 89.5 | 11.5 | 100 | 8.2 |
| | 2 | 100 | 100 | 100 | 100 | 100 | 100 | 100 | 11.1 | 100 | 8.2 |
| | 4 | 100 | 100 | 100 | 100 | 100 | 100 | 100 | 11.0 | 100 | 9.3 |
| Selective | 1 | 98.6 | 92.5 | 100 | 100 | 72.9 | 53.0 | 100 | 99.3 | 99.9 | 98.6 |
| | 2 | 99.3 | 93.7 | 100 | 100 | 91.3 | 66.7 | 100 | 94.8 | 100 | 99.9 |
| | 4 | 97.3 | 81.9 | 100 | 100 | 98.2 | 67.2 | 100 | 96.7 | 100 | 97.7 |
| Selective + Modular (mod4) | 1 | 100 | 100 | 100 | 100 | 86.1 | 27.3 | 58.1 | 27.4 | 98.5 | 83.1 |
| | 2 | 100 | 100 | 100 | 100 | 91.0 | 31.4 | 99.1 | 55.0 | 99.2 | 56.5 |
| | 4 | 100 | 100 | 100 | 100 | 99.2 | 35.9 | 100 | 57.9 | 100 | 55.1 |

Table A3: Results for recurrent architectures corresponding to Table A1, switching from best to **median** performance out of 5 runs. Bold entries indicate a >10 absolute gap between the median and best results.

## C  IMPORTANCE OF BOS FOR NoPE AND RoPE

We observed grevious failures of NoPE and RoPE on Counting + Succession and Modular counting (Table 2 rows 1 & 5), where both IND and OOD performances were below chance-level. The inability to fit in-domain data took us by surprise. We perform further analysis and find that NoPE and RoPE crucially depend on the presence of a begin-of-sequence (bos) token in order to fit the in-domain data. Concretely, we insert <bos>, a token distinct from the remaining vocabulary, at the beginning of each training sequence. Loss computation at the output side of <bos> is skipped. After such adjustment, NoPE and RoPE are able to achieve perfect IND accuracy on both Counting + Succession and Modular counting. Notably, <bos> also improves RoPE-OOD performance on modular counting well beyond the chance level. A question immediately arises: why other tasks were accomplished initially? To account for the IND success of NoPE and RoPE in other tasks studied in Section 4, we believe that the first token in a shifted start sequence may have concurrently played the role of <bos>[5]. For selective counting, the task nature may remove the dependence on bos.

The special role of bos has been discussed in the literature. Chiang & Cholak (2022) manually constructed Transformer weights for recognizing two regular languages, where bos plays key roles in multiple steps of their construction. Elhage et al. (2021) pointed out a phenomenon called "resting", where an attention head predominantly attends to a meaningless token by default (i.e. resting position) if there is no token matches what it looks for. The bos token and punctuation tokens are common examples of resting positions in a pretrained Transformer. Similarly, Vig & Belinkov (2019) discovered that 57% of attention was directed to the first token in pretrained GPT-2. These works provide clues for why bos is critical for NoPE and RoPE in certain counting tasks. A detailed answer is left for future work. Another intriguing question is why other PEs performed well without bos. One possibility is that different PEs tend to direct the Transformer towards different solutions, among which some relies on bos while others do not, implying a diverse algorithmic phase space (Zhong et al., 2024) in Transformers. Future work is needed to 1) characterize Transformers' algorithmic phase space, and 2) analyze how PEs constrain the "navigation" through such a space.

---

[5] For the Helper Token task, we reused the data generation process for shifted start sequences, simply fixing the shift to be 0. We realized later that this inadvertently offered the opportunity for the shifted start token to serve as <bos> in Helper Token task in the same way as in the Shifted Start task.

| Task | L | w/o bos | | | | w/ bos | | | |
|------|---|------|------|------|------|------|------|------|------|
| | | **NoPE** | | **RoPE** | | **NoPE** | | **RoPE** | |
| | | IND | OOD | IND | OOD | IND | OOD | IND | OOD |
| Vanilla | 1 | 2.0 | 0.0 | 2.0 | 0.0 | 100 | 0.0 | 100 | 0.0 |
| + Succession | 2 | 2.0 | 0.0 | 2.0 | 0.0 | 100 | 0.0 | 100 | 0.0 |
| | 4 | 2.0 | 0.0 | 2.0 | 0.0 | 100 | 0.0 | 100 | 0.0 |
| Modular (mod10) | 1 | 11.8 | 11.8 | 11.8 | 11.8 | 100 | 10.1 | 100 | 27.8 |
| | 2 | 12.0 | 11.8 | 11.8 | 11.8 | 100 | 8.6 | 100 | 40.3 |
| | 4 | 11.8 | 11.8 | 11.8 | 11.8 | 100 | 8.6 | 100 | 41.6 |

Table A4: Grevious in-domain failures of NoPE and RoPE can be addressed by the bos token, though their OOD performance benefits less from bos. This finding calls for future work to investigate how bos affects NoPE and RoPE, and why other PEs/tasks are not affected.

## D    MODULAR COUNTING ANALYSIS

This section complements Section 5. We describe the additional experiments and analysis performed upon in-depth investigation of what influential factors account for the empirical observations on Modular Counting. The mechanism permitting generalization to longer sequences consists of two steps: *First Token Recognition* and *Position-based Modular Subtraction*. These two steps demand a model to break symmetry and build representations indicative of PID%10, respectively. Figure A1 and Figure A4 show that the model's inner workings highly align with our mechanism.

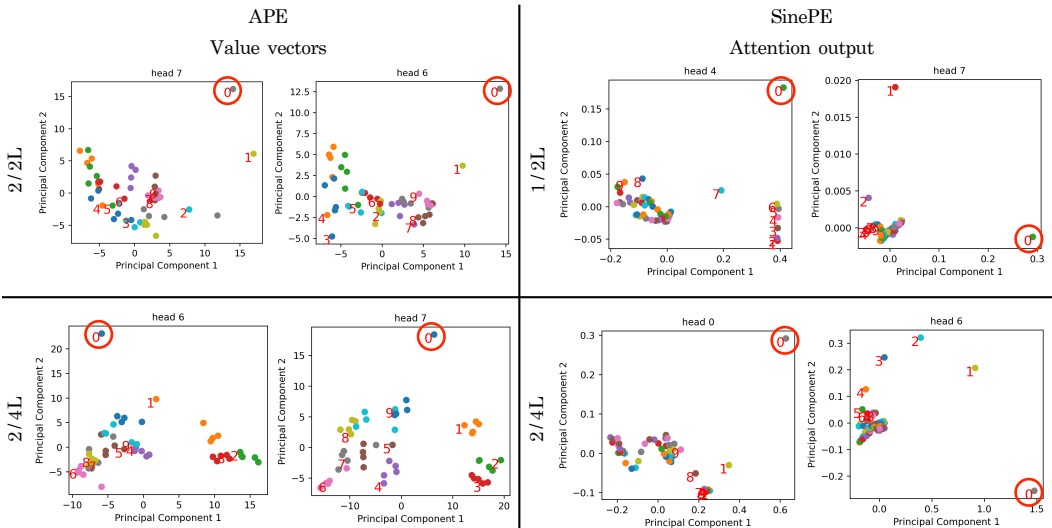

Figure A1: The first token's intermediate representations stand out w.r.t the first and second principal components. And this precedes the attention layer exhibiting an attention pattern concentrating on the first_tok.

| Model Configs | NoPE | | Sine | | APE | | RoPE | | SPE | |
|---------------|------|------|------|------|------|------|------|------|------|------|
| | IND | OOD | IND | OOD | IND | OOD | IND | OOD | IND | OOD |
| 1L, 1h, 32d | 0 | 0 | 100 | 100 | 100 | 100 | 0 | 0 | 100 | 100 |
| 1L, 1h, 4d | 0 | 0 | 100 | 100 | 100 | 100 | 0 | 0 | 100 | 100 |

Table A5: Results of 1L Transformers trained on the ***first_tok_homogeneous*** task. We report the accuracy of only the first token's prediction, because a model would easily learn to predict the majority label 'F', which would already allows for a fairly high average accuracy.

First, we design a ***first_tok_homogeneous*** task to examine how well each PE schema supports the inductive bias for breaking the symmetry of a homogeneous sequence. The input-output format is

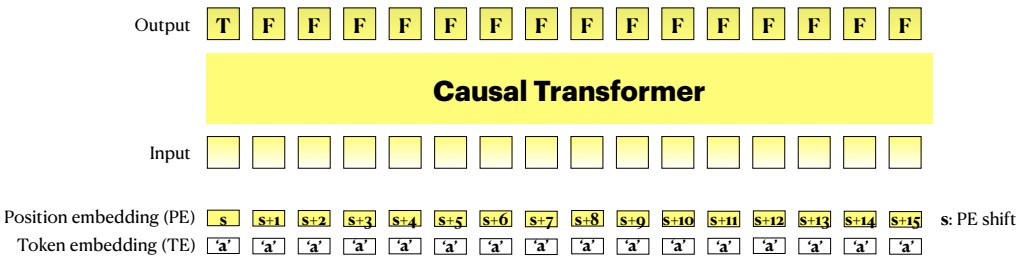

Figure A2: Input-output format of the ***first_tok_homogeneous*** task, designed for the purpose of examining which types of PE can be leveraged by Transformers to distinguish among identical tokens.

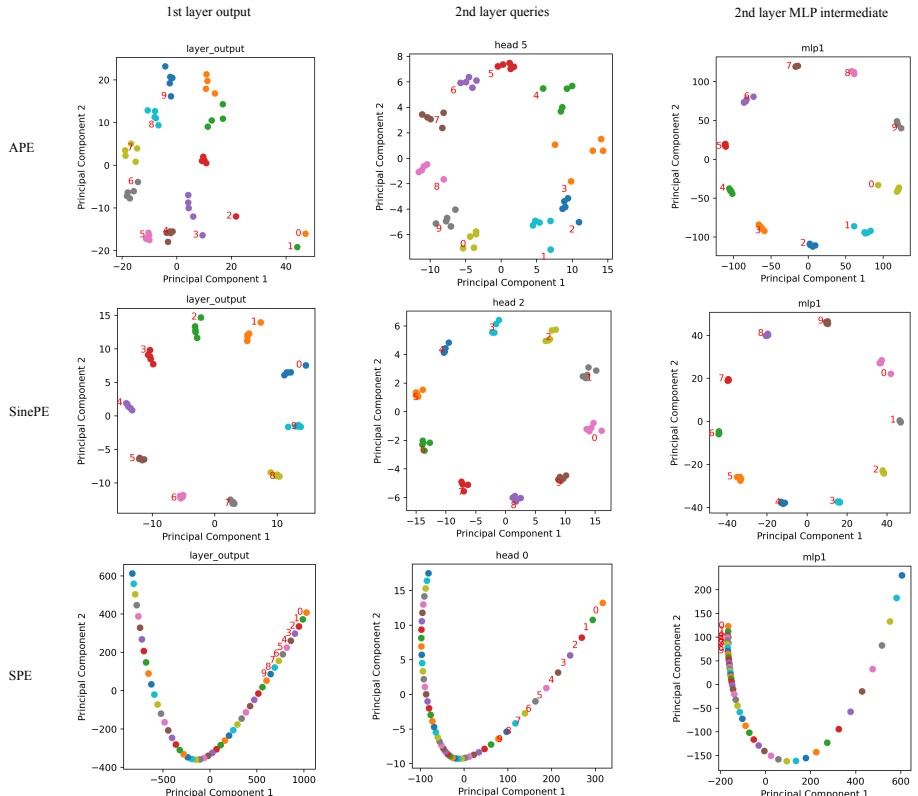

Figure A3: PCA of intermediate states for 2L APE/SinePE/SPE Transformers trained on modular counting. This differentiates SPE from APE/SinePE in the ability to count modularly, although all three PEs accomplish fist token recognition.

illustrated in Figure A2. Each token is supposed to predict a binary label indicating whether it is the first token in the input sequence. We construct training/IND sequences of length 50 and OOD sequences of length 128. Following the experimental setup in Section 3, we 1) adopt the position-shift augmentation, 2) run each training job up to 312.5K or 625K steps (depending on when the model exhibits strong signs of saturation or plateauing), 3) select the best checkpoint over the entire training course for each training job, and 4) report the best result out of five seeds. The metric is accuracy, for which we only evaluate on the first token's prediction. This is because a model would easily learn to predict the majority label 'F', which would already allows for a fairly high average accuracy. It is indeed observed that all models have perfectly learned to predict 'F' on non-first tokens, so the performance difference is only manifested via the first token's prediction. Table A5 presents the results: Transformers equipped with SinePE, APE or SPE can recognize the first token among a homogeneous sequence, while NoPE and RoPE cannot. It is worth explaining that RoPE fails

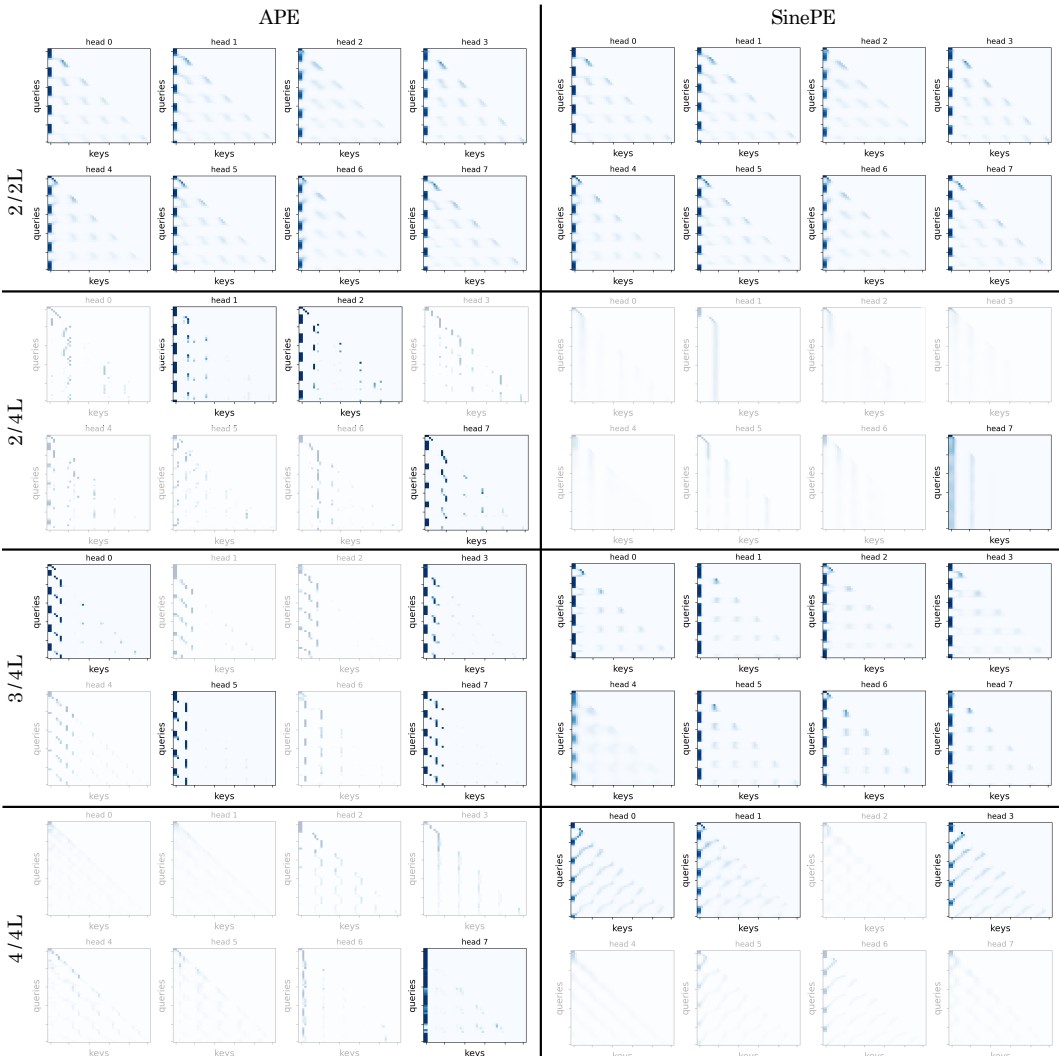

Figure A4: Heads that concentrate their attention on the first_tok are prevelant beyond the first layer. Note that first_tok heads do not unanimously attend the to the first_tok. Instead, they are selective for position indices with specific mod-10 values. Encoding such fourier features allows multiple first_tok heads to collaboratively attend to the first token and may help subsequent layers in performing modular arithmetic.

because it only modifies queries and keys, leaving the values identical for a homogeneous sequence. No matter how the attention weights vary, the attention output (i.e. weighted sum of identical value vectors) would still be identical throughout the sequence.

Second, to investigate whether a model builds representations indicative of PID%10, we visualize principal components of intermediate states for 2L APE/SinePE/SPE Transformers trained on modular counting. The fact that APE/SinePE models construct 10 dinstinct groups of representation whereas SPE models do not, differentiates SPE from APE/SinePE in the ability to count modularly, although all three PEs accomplish fist token recognition.

Concretely, we feed an input sequence of 128 'a' tokens with PIDs 0-127 and record representations at the **1st layer's output**, the **2nd layer's queries** and the **2nd layer's MLP intermediate states**. In Figure A3 we plot the first two principal components for 128 tokens, color coded by PID%10, e.g. tokens whose PIDs belong to 1, 11, 21, 31 ... are assigned the same color. To implement a generalizable solution for modular counting, tokens with the same value of PID%10 should have close representations. From the plot we see that the APE/SinePE representations form clean clusters based on PID modulo 10, while the SPE representations do not.

| | w/ Causal Masking | | w/o Causal Masking | |
|---|---|---|---|---|
| Model Configs | IND | OOD | IND | OOD |
| NoPE 1L, 1h, 4d | 100 | 100 | 4.0 | 0.0 |

Table A6: Results of 1L Transformers trained on the ***first_tok_heterogeneous*** task. We report the accuracy of only the first token's prediction, because a model would easily learn to predict the majority label 'F', which would already allows for a fairly high average accuracy. PEs are not implemented in order to demonstrate that Transformers can simply leverage causal masking to accomplish this task.

## E   SELECTIVE COUNTING ANALYSIS

This section complements Section 5. We describe the additional experiments and analysis performed upon in-depth investigation of what influential factors account for the empirical observations on Selective Counting. Section 5 proposes two factors demanded by a mechanism that permits generalization: *First Token Recognition* and *Token-based Attention*. Unlike modular counting, where the inputs are homogeneous, selective counting is designed to have heterogeneous inputs containing 10 unique tokens. *First Token Recognition* on heterogeneous inputs can be achieved purely with causal masking. PEs may aid *First Token Recognition*, but at the meantime introduce intricacies interfering with *Token-based Attention*, as *Token-based Attention* ideally requires for the assignment of attention weights irrespective of positional information.

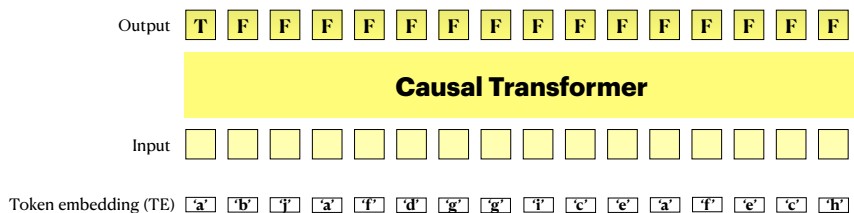

Figure A5: Input-output format of the ***first_tok_heterogeneous*** task, designed to demonstrate that a **causal** Transformer can recognize the first token in a **heterogeneous** input sequence without the need for PEs.

First, we design a ***first_tok_heterogeneous*** task to demonstrate that causal masking enables *First Token Recognition* on heterogeneous inputs. The input-output format is illustrated in Figure A5. It differs from ***first_tok_homogeneous*** in that 1) each input sequence contains a random number of 10 different tokens, and 2) the removal of PEs. We construct training/IND sequences of length 50 and OOD sequences of length 128, randomly selecting the number of occurrences for each token and randomly shuffling the sequence. The experimental setup and evlauation metric are identical to ***first_tok_homogeneous***. Table A6 presents the results: With causal masking, a NoPE Transformer can accomplish the task with as few as one head and four hidden dimensions. This is no longer achievable without causal masking, emphasizing the critical role played by causal masking in ***first_tok_heterogeneous***. In fact, we are able to understand the inner workings of a causal Transformer on ***first_tok_heterogeneous***. Thanks to causal masking, the first tok will have nothing to attend to other than itself. Non-first tokens will merge features from predecessors during attention. MLP layers will learn to distinguish between representations resulted from merges vs. representations that have not been merged.

Second, we argue that PEs complicate the picture, because PEs might be initially involved in *First Token Recognition*, but should better be ignored in order for *Token-based Attention*. This explains why NoPE 2L generalizes well in selective counting, whereas APE/SinePE/SPE only achieves generalization with 4L (Table 2 row 6). SPE's performance is second to NoPE in a sense that it suffers less from the complication introduced by PEs — only a single dim is affected by PIDs in SPE.

The complication caused by PEs is further evidenced through comparing performance with vs. without BOS. Concretely, we repeat our selective counting experiments except for the explicit inclusion of a beginning-of-sequence (BOS) token. With BOS, PEs need not to be actively engaged in *First Token Recognition*. Hence, APE and SinePE will receive cleaner supervision signals encouraging them to

| Task | L | NoPE IND | NoPE OOD | Sine IND | Sine OOD | APE IND | APE OOD | RoPE IND | RoPE OOD | SPE IND | SPE OOD |
|---|---|---|---|---|---|---|---|---|---|---|---|
| Selective | 1 | 100 | 9.3 | 100 | 68.8 | 100 | 10.6 | 100 | 29.9 | 100 | 61.3 |
| Selective + bos | 1 | 100 | 99.5 | 100 | 98.3 | 100 | 99.4 | 100 | 68.2 | 100 | 98.3 |
| Selective | 2 | 100 | 94.1 | 100 | 32.6 | 100 | 13.9 | 100 | 49.1 | 100 | 86.9 |
| Selective + bos | 2 | 100 | 99.9 | 100 | 100 | 100 | 95.8 | 100 | 40.9 | 100 | 99.9 |

Table A7: Selective Counting requires a subtask of first token recognition. Such a requirement can be removed by explicitly providing a BOS. With BOS, NoPE and SPE Transformers can learn a generalizable solution using one layer less than what would be required without BOS. Moreover, the once noticeable performance gap between APE/SinePE and NoPE disappears with BOS.

ignore PEs. As expected, the performance gap between NoPE/SPE and APE/SinePE vanishes when BOS is explictly included — they generalize equally well on selective counting with either 1L or 2L (Table A7).

Lastly, we explain why RoPE is inferior to other PE schemas in both Selective and Selective+BOS. We hypothesize that the enforcement of recency bias in RoPE makes it hard to escape the influence of positions when computing attention weights. This is in contrast to APE/SinePE/SPE in which case it is possible to ignore the influence of PEs through learning. To measure the amount of sensitivity to PEs in the computation of attention weights, for each pair of $(\mathrm{TID}_{\mathrm{source}}, \mathrm{TID}_{\mathrm{target}})$, we compute the std of $q_{\mathrm{source}} k_{\mathrm{target}}^T$ across all $(\mathrm{PID}_{\mathrm{source}}, \mathrm{PID}_{\mathrm{target}})$ pairs that the source and target tokens can possibly take. Figure A6 visualizes such variation in attention weights for 1L APE/SinePE/SPE/RoPE Transformers trained on the Selective+BOS task, color coded according to the std value interval. Darker cells suggest larger std values, implying greater amounts of variation in attention scores caused by PEs. Attention scores in an APE model are the least affected by varying PIDs, which is conducive to *Token-based Attention*. On the other hand, a RoPE model exhibits the greatest sensitivity to PEs when computing attention weights, suggesting that it insufficiently learns to disregard PEs as required by the task. Such a shortcoming of RoPE prevents it from generalizing on selective counting.

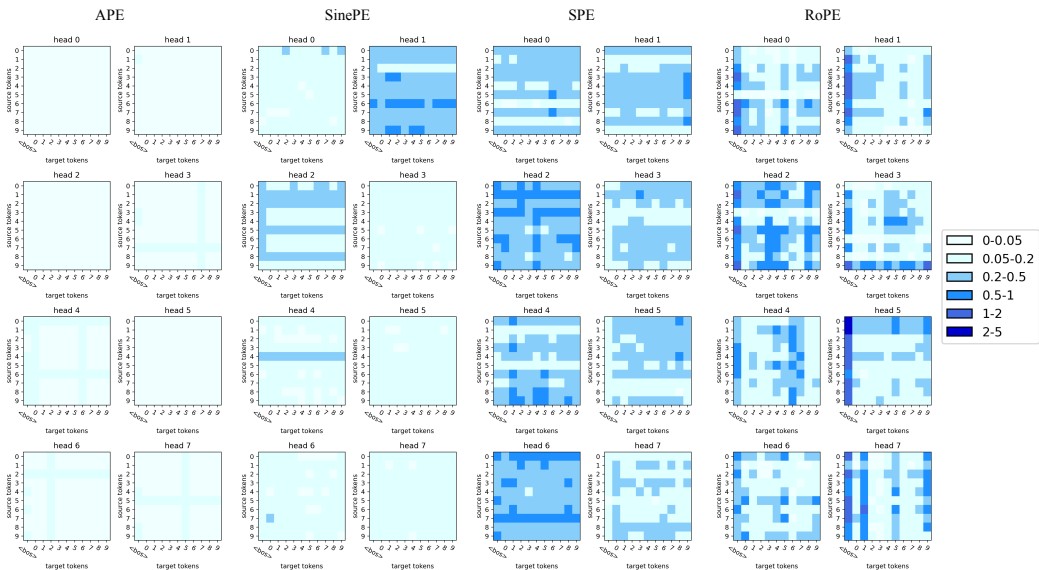

Figure A6: Variation in attention scores influenced by PEs. First, we take the best performing 1L checkpoints trained on the Selective Counting + BOS task and compute attention scores (pre-softmax $qk^T$) between all pairs of tokens. Then we compute the std of attention scores across all possible assignments of PIDs. Hence, the value of each cell in each of the panels visualized here equals to the std of $q_{\mathrm{source}} k_{\mathrm{target}}^T$ values across all PID pairs that the source and target tokens can take (i.e. $\mathrm{PID}_{\mathrm{source}} \geq \mathrm{PID}_{\mathrm{target}}$ following the causal rule).

# F ADDITIONAL RELATED WORKS

## F.1 FORMALLY CHARACTERIZING TRANSFORMER EXPRESSIVITY

RASP (Weiss et al., 2021) introduces a functional library of Transformer capabilities, based on the Python programming language. It lays an important foundation for follow-up works to expand on. For example, Tracr (Lindner et al., 2023) proposes a compiler that allows for automatic crafting of Transformer weights from a RASP script. This facilitates the comparison of solutions learned through SGD versus "canonical" ones a human would create. Zhou et al. (2023) updates the RASP library by adding more realistic assumptions on the numerical bounds, as well as how the position indexes can contribute to the computation graph.

Others characterize Transformer computations from the perspective of boolean circuits (Cong et al., 1996; Yang & Chiang, 2024; Strobl et al., 2024; Merrill et al., 2022), where primitives are atomic boolean operations. This is distinct from RASP, Tracr and our work, as we deal with functional primitives, but may contribute complementary insights. Also note that there lacks empirical validations accompanying this line of research, opening the room for future work to bridge this gap.

Transformers can be analyzed through Automata Theory (Liu et al., 2022; Del'etang et al., 2022). This branch has not reached agreement on where the hard boundary for Transformer's capacity lies. For example, while (Hahn, 2020) proves the hard limits of self-attention on modeling Dyck, (Yao et al., 2021; Ebrahimi et al., 2020) demonstrated empirical success. We believe that nuances regarding the task format, architectural modifications and the definition of generalization vary across studies. It invites significant future contributions to either remove this complication or unify these nuances.

## F.2 RECOGNIZING THE PARITY LANGUAGE

There is a strong connection between our work and previous works on PARITY or counter languages. Hahn (2020) suggested that Transformers have theoretical difficulty in expressing PARITY, while Chiang & Cholak (2022) demonstrated that such limitation can be overcome via hand-constructed weights Regarding this topic, it is important to distinguish between two notions of difficulty: difficulty to fit in-domain data vs. difficulty to length-generalize. In the context of theoretical discussions, the difficulty of PARITY for transformers often means the inability to fit the training data when the training sequence length approaches infinity, i.e. Transformers cannot *express* PARITY for arbitrarily long sequences. Such a limitation is empirically corroborated for pure-attention Transformers. Bhattamishra et al. (2020) argued for the necessity of PEs in recognizing PARITY, while admitting that PEs learned in-domain cannot be used for sequences of higher lengths. Our results similarly suggest that PEs enable Transformers to fit the training data where NoPE fails to learn. However, "difficulty" in the context of this paper mainly refers to generalization beyond the training SEQLEN range. The PE shift trick used in our experiments presents one step towards better generalization performance. But we demonstrate that PE shift only lifts one barrier towards generalization (OOD-position), leaving the other barriers (OOD-cardinality, OOD-range-of-dependency) unsolved.

## F.3 RECURRENCE WITH ATTENTION

Many studies attempt to promote new architectures that address the fundamental tradeoff between efficient training and efficient inference. Transformers achieve the former thanks to the parallel attention mechanism, while an RNN features the latter with linear time complexity and constant history memory. Towards achieving the best of both worlds, previous works have focused on either proposing attention-ish mechanisms with subquadratic time complexity (Zhai et al., 2021; Peng et al., 2023; 2024), or modernizing RNN to permit parallelized training (Gu et al., 2021b; Gu & Dao, 2023; Nguyen et al., 2024). We refer the reader to Tiezzi et al. (2024); Wan et al. (2023) for surveys on this emergent area. Extending the expressivity analysis for Transformers to these new RNN architectures (Merrill, 2019; Merrill et al., 2024) would be a fruitful direction.

# G ADDITIONAL FIGURES AND TABLES

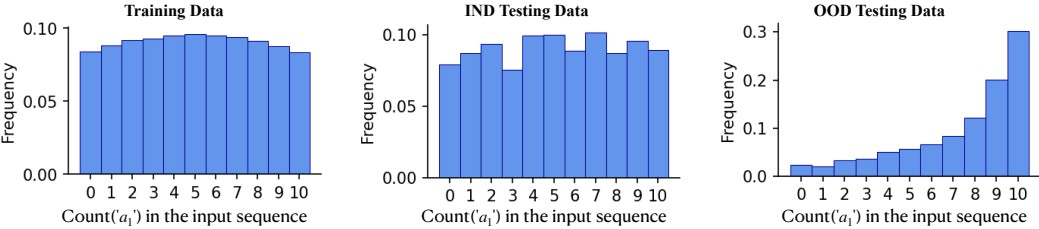

Figure A7: Distribution of the number of occurrences for token '$a_1$' in Selective Counting. In the OOD test set, Count('$a_1$') is skewed towards larger values because OOD sequences are longer while we restrict each token to appear less than ten times to avoid OOD-cardinality. Tokens '$a_2...a_{10}$' have similar distributions.

| | | MAX_TRAIN_SEQLEN \| MAX_OOD_SEQLEN | | | | | | | | |
| | | 25 \| 50 | | 50 \| 100 | | 25 \| 100 | | 50 \| 200 | | 25 \| 200 | |
| Task | L | IND | OOD | IND | OOD | IND | OOD | IND | OOD | IND | OOD |
|---|---|---|---|---|---|---|---|---|---|---|---|
| Shifted Start | 4 | 100 | 84.3 | 100 | 98.9 | 100 | 30.9 | 100 | 22.1 | 100 | 11.4 |

Table A8: Performance of RoPE 4L Transformers on Shifted Start Counting with varied combinations of MAX_TRAIN_SEQLEN and MAX_OOD_SEQLEN. The MAX_PID hyperparameter is also adjusted in proportion to MAX_OOD_SEQLEN. Generalization deteriorates as the ratio between MAX_OOD_SEQLEN and MAX_TRAIN_SEQLEN becomes greater.

