# OpenReview forum: "Language Models Need Inductive Biases to Count Inductively"
_ICLR.cc/2025/Conference — ICLR 2025 Poster_

### Official Review · Reviewer_FunV · 2024-11-01

**Soundness:** 4
**Presentation:** 4
**Contribution:** 4
**Rating:** 8
**Confidence:** 4

**Summary:**

This paper questions the common belief that counting is a basic, easy-to-learn skill of transformer-based language models. It provides a detailed study of what it means for a language model to learn to count and the conditions that affect its ability to do so. The authors set up different counting tasks of varying levels of difficulty to test when and how transformers can learn to count. They compare different types of transformer models (mainly those with different positional encodings) and include comparisons to classic and modern RNNs. Results show that counting isn’t a natural skill for transformer and modern RNN models, especially when they need to generalize to counts they haven’t seen in training—a key requirement for inductive counting.

**Strengths:**

The paper is well-written, organized, and easy to follow. The authors back up their claims with strong evidence and well-designed experiments, and they clearly explain the purpose and outcomes of each experiment. In particular, I like that:
- The paper questions a basic assumption and shows that counting isn’t necessarily something language models can be expected to master but instead something they have to learn under specific conditions. I find this to be an important and illuminating insight.
- By testing various transformers with different positional encodings and comparing them to RNNs, the paper provides a broad picture of how different architectures handle counting.
- The finding that counting isn’t a built-in skill could shape future work in language model design and applications, as it highlights the need to consider how well a model can generalize basic counting tasks.

**Weaknesses:**

I couldn't find any major weaknesses in the paper. The one small issue I ran into was with the term “number word,” which wasn’t immediately clear. Adding an example early on—like explaining that you use decimal numbers as “number words”—could help readers follow this part more easily.

**Questions:**

I don’t have any further questions. The paper does a thorough job of explaining its goals and findings.

---

> ### Author Response · Authors · 2024-11-20
> **Thank you so much for your recognition of our contribution and deep engagement in reviewing this work.**
>
> **1. The meaning of “number word” isn’t immediately clear**
>
> We will clarify this borrowed terminology. In cogsci, it is proposed that children initially develop mental representations for “sets of cardinality k”, distinctly from how they represent number words (the meaning they attribute to the utterances of “one”, “two”, “three” …).
>
> Correspondingly, in the LM context, number words are connected to how a model embed linguistic symbols “one”, “two”, “three”, which may differ from how a model represents a set of tokens with cardinality k.

---

> > ### Comment · Reviewer_FunV · 2024-11-24
> >
> > Thank you for the clarification! I maintain my positive evaluation of the work.

---

### Official Review · Reviewer_RT8E · 2024-11-02

**Soundness:** 3
**Presentation:** 2
**Contribution:** 2
**Rating:** 5
**Confidence:** 4

**Summary:**

The authors study the effect that the choice of position encoding in transformers has on their length generalization for counting. Specifically, they train transformers (<= 4 layers) on various counting problems, and evaluate their OOD performance. The counting problems require transformers to output a token representing the number of occurrences of certain input tokens. The variants of counting include shifted start, where the first input token is an offset to the counts transformers should output, modular arithmetic, and selective, where multiple classes of tokens are present in the input, and the counts of each class are to be tracked independently. Overall they find differences between the five position embeddings studied (NoPE, Sine, APE, ROPE and SPE), and conduct deeper analyses in the appendices to explain some of the more surprising findings.

**Strengths:**

- Comprehensive empirical evaluation of the 5 position embeddings.
- Analysis is reasonably thorough. e.g. I like the finding that RoPE fails to do modular counting, but not if there is a BOS token.

**Weaknesses:**

Counting setting far removed from practical language models.


Issues with overclaiming in the interpretation and discussion of results. Specifically:
- "poor results for 1L and 2L models suggest that counting in Transformers may require a non-trivial computation budget". I do not think the results are strong enough to support this claim. Firstly, 4 layers are still only a small fraction of most practical language models (e.g. even llama 8B has 32 layers), so "non-trivial" is some what of a stretch. Secondly, this studies the "inductive bias", i.e. the outcome of the particular training dynamics on this toy problem; it's entirely possible that some other synthetic set up, or even pretrained language models, can be more efficient. Claims like this is better substantiated with interpretability on pretrained language models, or an expressivity argument
- "Our results motivate the integration of multiple PE schemas to take advantage of orthongonal strengths." At no point do you study the setting of integrating several PEs. In fact, your results seem to suggest the opposite: that sometimes having PE with more degrees of freedom can do worse than having fewer (e.g. how NoPE can do better at some tasks)

**Questions:**

- Do you think that expressing numbers in bases (in the sense of base 10) is a source of inductive bias that can help length generalization? Would your results differ significantly?
- Can you draw high level lessons from this? How would the findings here help improve the design of position embeddings?

---

> ### Author Response · Authors · 2024-11-20
> **Thank you! We hope our reponses help address your concerns about the interpretation and practical value of our results.**
>
> **1. Counting setting far removed from practical language models.**
>
> We respectfully disagree with this statement. We may not directly conduct experiments with practical LMs, but there is deep connection between the subject of our investigation and practical scenarios. Such deep connection has been well argued in previous literature. For example, the formal expressivity community has dedicated studies on counting [1] and recognition of counter languages [2,3] with Transformers/self-attention. The LLM community recognizes the need for counting the depth when reasoning through a structured problem space such as graph [4], tree [5], recursive function call [6] or dynamic programming [7]. For a rigorous study, it is necessary to abstract away from the practical scenario in order to peel off confounding factors and make it possible to develop informed and justified arguments.
>
> [1] Yang & Chiang, “Counting like transformers: Compiling temporal counting logic into softmax transformers”
>
> [2] Ebrahimi, et al., “How can self-attention networks recognize dyck-n languages?” Related to numerical reasoning or number sense (borrowed from cognitive science).
>
> [3] Bhattamishra, et al., “On the ability of self-attention networks to recognize counter languages”
>
> [4] Dziri, et al., “Faith and fate: Limits of transformers on compositionality”
>
> [5] Zhu &  Li., “Physics of language models: Part 1, context-free grammar”
>
> [6] Zhang, et al., “Can transformers learn to solve problems recursively?”
>
> [7] Del’etang, et al., “Neural networks and the chomsky hierarchy”

---

> ### Author Response · Authors · 2024-11-20
>
> **2. Issues with overclaiming in the interpretation and discussion of results.**
>
> **2.1 Counting requires a “non-trivial budget” is an overclaim**
>
> Discussion in this respect might depend on what is regarded as “non-trivial”. Our work begins with questioning the assumption that counting is an atomic function in the formal computational framework of Transformers. An atomic function should be effortlessly implemented or learned, because they are expected to be repeated and composed for tens or hundreds of times throughout a deep model. It is in this regard that we define “trivial”. Thus, a 4L-worth of computation w.r.t to a 32L model is non-trivial, because the model cannot afford repeating and composing a 4L-worth of computation for tens or hundreds of times, let alone interleaving it with other non-trivial computations once a model is expected to be versatile. We are effectively borrowing the scoping of non-trivial from the implicit assumptions already in the literature, it is not a subjective judgement on our part, but we will refine the discussion further to remind the reader of this basis.
>
> **2.2 This studies the "inductive bias", i.e. the outcome of the particular training dynamics on this toy problem; it's entirely possible that some other synthetic set up, or even pretrained language models, can be more efficient. Claims like this is better substantiated with interpretability on pretrained language models, or an expressivity argument**
>
> **`We apologize if we have misunderstood this question but are having difficulty understanding the claim.`**
>
> We assume, the reviewer is not arguing that pretraining with autoregressive losses can imbue a model architecture with new expressivity, thereby shifting its position on the computational hierarchy?  So we attempt to outline answers for other interpretations below but would really appreciate it if the reviewer could elaborate or decompose the question, as we have clearly not fully understood their request and would like to!
>
> - We agree that inductive bias can have multiple sources: the architecture, training curriculum, the gradient descent procedure, to name a few. Analyzing multiple sources of inductive bias altogether can either reveal a broader picture, or ***obscure*** it. So one has to be careful when navigating multiple factors at play. It is for this reason that we restrict our focus on the inductive bias encoded in the ***architecture*** while ***fixing the task formatting*** and ***training procedures***. It is true that our results cannot say much about the role played by factors beyond the architecture. But what we are able to say about the architecture, is well grounded and justified through experiments. This is a major contribution and it is unfair to diminish this contribution based on the absence of parallel investigation on task formatting and training procedures.
> - The reviewer suggested that we leverage pretrained LMs or other synthetic setups to achieve greater efficiency, which is confusing because our goal is not to propose more efficient approaches on counting, but to provide a broad picture of how current approaches handle counting. Better understandings of where we currently are is always an essential step towards making informed proposals about next steps.
> - Interpretability and expressivity are neighboring fields which have common motivation and research questions with this paper. However, it is infeasible to span all these fields in a single paper. We believe our work will interest audiences in both interpretability and expressivity, thereby worthing publication for it to reach its audiences.
>
> **2.3 The paper suggested the integration of several PEs without actual experiments. “In fact, your results seem to suggest the opposite: that sometimes having PE with more degrees of freedom can do worse than having fewer”**
>
> Thanks for the question. We did not conduct these additional experiments because the potential benefits of integrating PEs is an ***implication of the main findings***, rather than the main argument itself. The actual operationalization of this idea, however, requires nontrivial effort and may become a full paper in its own right. As the reviewer has already noticed, our results suggest both the orthogonal strengths of multiple PEs, ***AND*** the interference PEs can cause when incorporated untimely. Therefore, harnessing the orthogonal strengths requires a more organic integration beyond plug-and-play, in order not to suffer from interference. We believe there is a large future venue for such investigation that is beyond the scope of a single paper. But the fact that our findings open up new questions around PEs, and direct research discussions to these questions, justifies the importance of this work.

---

> > ### Comment · Reviewer_RT8E · 2024-11-23
> >
> > Thanks for the response.
> >
> > My primary remaining concern is the inability to form strong, generalizable, and useful conclusions from this work. Part of this is that the work is limited by the methodology of training toy transformers on toy datasets and studying their performance. The conclusions drawn here are thus contingent simultaneously on the toy architecture, the toy task, and the toy training procedure.
> >
> > In contrast, suppose one undertakes a interpretability approach of studying pretrained LMs on toy counting tasks. This uses a realistic architecture, trained in a realistic setting, but for ease of analysis uses a toy task. Lessons drawn from interpretability research can plausibly generalize to more complex tasks.
> >
> > Suppose instead one makes an expressibility argument that argues that a particular transformer variant can or cannot perform a certain toy task. Again, this analyzes a realistic architecture, and because expressibility is a statement that applies to all possible weights, conclusions will apply to pretrained LMs as well.
> >
> > Unfortunately, this work does neither, and instead trains a lot of small transformers on small tasks, and evaluate them on toy counting tasks. This makes it really challenging to generalize their findings.
> >
> > Unless the authors can demonstrate some way that their findings can have implications for real models, it is difficult for me to increase my score. For example, one way to do this is to exhibit qualitatively similar effects in pretrained LMs. Another way is to train models on more realistic data distributions and show that the training dynamics on the more realistic distribution will lead to different kinds of generalization properties when the model has different position encodings.

---

> > > ### Author Response · Authors · 2024-11-23
> > > **Thank you for engaging -- misunderstanding of how data affects formal expressivity?**
> > >
> > > Thank you for engaging with our work! We are however concerned that we seem to be talking past one another. It sounds like you're proposing that factors like data scale would change our results or that "real" models are somehow different than what we are discussing here. For simplicity, I will borrow the terminology from theory of computation to help get us aligned.
> > >
> > > Assume you have a model whose expressivity is bounded to regular languages (formal definition of regular) and whose recognizing automata is therefore finite-state. If data is presented to learn that FSA from a context-free or context-sensitive language (e.g. one requiring a PDA or linear-bounded non-deterministic Turing machine), provably, there is no amount of data or parameters (states) that would allow the FSA to cover the phenomena in the data or generalize to the full space of the language.  This is a fundamental truth of computation.
> > >
> > > It's possible, that you're arguing that transformers are Turing complete (aka capable of expressing recursively enumerable language).  But as you know, that is not an established fact.  There are some works [Perez et al. ICLR 2019, JMLR 22] that try to scope those claims but require key assumptions (e.g. arbitrary precision, choices of functions, etc).  While their result is beautiful, it does not claim that the transformers in use today are Turing complete nor that such a model is learnable. In fact, to study the question of the learnability of recursive enumeration would require a controlled setting whose base case can be systematically probed. Which is, precisely what is being presented in our work! So I think, your argument is that real world scenarios require evaluating the model architecture on languages that are not FSA verifiable and that demonstrating failure in this case has implications for all transformer based experiments -- and that is precisely where we are in agreement and is the core contribution of our work.

---

> ### Author Response · Authors · 2024-11-20
>
> **3. Why not using base10 representation format of numbers?**
>
> Base 10 number representation would lead to much more difficulty in experimental design. When a number could be tokenized into multiple tokens, we could no longer elicit a single-token output for each cardinality, demanding pre- and post-processing to resolve the one-to-many alignment.
>
> **3.1 Will base10 help length generalization?**
>
> As opposed to bringing help, base10 may introduce unnecessary intricacies, i.e. the mapping back and forth between the concept of a number as a whole and its representation by multiple tokens w.r.t an LM’s vocabulary.
>
> To elaborate on this intricacy, previous studies show that when a word is tokenized into multiple tokens, early Transformer layers serve to "detokenize" —— mapping token spans into internally-represented words, while late Transformer layers take additional reponsibility to "re-tokenize" —— mapping internally-represented words back to token spans. [1, 2]. The implication of these studies on counting is that whole-numbers are preferred as the processing unit in Transformers, while tokenization adds extra complexity of mapping token spans and numbers back and forth. In our setting, we use whole-number tokenization to ***avoid conflating the complexity of counting with the complexity introduced by tokenization*** which is orthogonal to our focus.
>
> [1] Section 6.3.2-6.3.3 in Elhage, et al., "Softmax Linear Units", Transformer Circuits Thread
>
> [2] Feucht, et al., "Token Erasure as a Footprint of Implicit Vocabulary Items in LLMs”
>
> **3.2 Would base10 yield significantly different results?**
>
> Base 10 number representation may remove OOD vocabulary, but ***it only superficially dissolves*** the difficulty of counting. As explained in Line88-90, there is an important distinction between numbers in the ***language*** context and numbers in the ***cardinal*** context. ***Unseen cardinality values*** ***will persist with*** base10 representation. The reviewer might have missed an important point that OOD extrapolation is hindered by multiple types of OOD issues, including but not limited to: OOD-position, OOD-vocabulary, OOD-cardinality, OOD-range-of-dependency. As pointed out in Line457-460, these issues shall not be confused with each other and shall be targeted by separate remedial techniques. While base10 representation might remove the OOD-vocabulary issue, it leaves other issues unsolved. For this reason, we believe the results ***wouldn’t*** differ significantly if we switched to base10 representation.
>
> **4. Can you draw high level lessons from this?**
>
> Our work offers two high-level messages: 1) counting is not a built-in skill in most LM architectures, 2) counting could be imbued via certain architectural components that encode the relevant inductive bias. The first message calls for the field to reexamine what types of computation are assumed to be atomic. The second message suggests a potential direction for ***acquiring*** basic counting skills now that they cannot be taken for granted.
>
> **5. How would the findings here help improve the design of position embeddings?**
>
> Please see response to 2.3. We do not have concrete ideas at present, but we point out important considerations should a future study formalizes concrete ideas.

---

### Official Review · Reviewer_oPHk · 2024-11-03

**Soundness:** 2
**Presentation:** 3
**Contribution:** 2
**Rating:** 6
**Confidence:** 5

**Summary:**

This work emphasizes the limitations in language models for counting tasks. The authors examine various architectures, including RNNs, Transformers, and modern state-space models with a suite of experiments. They find that while traditional RNNs can easily count inductively, Transformers struggle without additional design considerations, and modern state-space models face degraded performance.

**Strengths:**

1. This work focuses on a specific problem, the counting task, for the language models. The authors conduct many experiments to investigate the ability to count systematically.
2. This paper not only focuses on the standard transformer architectures but also investigates many popular modern architectures.

**Weaknesses:**

1. **Lack of Insights:** Although this work conducts many experiments to support their findings, it offers limited insights into the reasons behind the poor performance of Transformers and modern RNNs. I encourage the authors to provide more intuition or explanations for the observed empirical phenomena.

2. **Lack of Generality:** This paper focuses on the counting task, which I acknowledge is an important task. However, it remains unclear how performance on this task influences real-world applications. The conclusions are specific to counting tasks and may not generalize well to real-world scenarios.

3. **Lack of Novelty:** While the paper addresses an important aspect of language model performance, it does not introduce significantly new concepts or methodologies. Similar studies[1] have already explored the limitations of Transformers in various tasks. This work just conducts some experiments in different architectures and does not offer sufficient contributions.

[1] When Can Transformers Count to n?

**Questions:**

How can the conclusions of this paper generalize to real-world tasks, such as math, code, and so on?

---

> ### Author Response · Authors · 2024-11-20
> **We seek to address misunderstandings and provide more information below.**
>
> **1. Lack of insights into the reasons behind the poor performance.**
>
> We apologize if Section 5 of the paper was too dense, but it is **`entirely devoted to the reasons behind the observed performance.`** To recap, we describe two generalizable mechanisms, for modular and selective counting, respectively. We provide evidence for why we believe the models are indeed implementing such mechanisms. Each mechanism demands particular inductive biases. Each PE schema either supports or goes against certain inductive biases. The high level explanation, put briefly, is that poor performance is resulted from the mismatch between a particular PE’s inductive bias and the task’s demand. Due to the space limit, we include detailed figures in Appendix D&E.  Again, we recognize there are many results and details which answer all of your concerns, but may not have been easy to digest. **We hope the reviewer is able to revisit those sections in the manuscript in more detail so we can have a technical discussion about the results.**
>
> **2. The conclusions are specific to counting tasks and may not generalize to real-world scenarios.**
>
> We believe the abstract concept of induction, lying at the core of inductive counting, is a commonality underlying a broad-enough range of real-world tasks.  In fact, the reviewer agreed on the importance of the counting task. We believe that such importance attached to counting exactly reflects the common recognition in the field that counting can be thoroughly studied under clean setting, while having implications outside the clean environment. It is unclear in this mathematical context what the review intends by “real world”, so precise examples of cognitively relevant mathematical phenomena would be greatly appreciated.

---

> > ### Comment · Reviewer_oPHk · 2024-11-21
> >
> > Thanks for the clarification.
> > This paper proposes two generalizable mechanisms, for modular and selective counting. For example, for modular counting, this paper introduces a mechanism that allows perfect generalization, which consists of two steps: First, token recognition and second, position-based modular subtraction.  This mechanism may be truly generalizable, but there may be other generalizable mechanisms. If the authors can provide more solid evidence that the transformer model really adopts this mechanism and not some other (e.g., the heat map of the attention score is consistent with your mechanism), this paper will provide more insight to the community to understand the transformer that performs counting.

---

> > > ### Author Response · Authors · 2024-11-24
> > > **This is a great follow-up suggestion! We have updated the manuscript with additional plots that support our mechanism.**
> > >
> > > We provide visualizations showing that the model's inner workings highly align with our mechanism
> > >
> > > Figure A1 ---- First, first token recognition is in line with the observation that the first token's intermediate representations **stand out w.r.t the first and second principal components**. And this precedes the attention layer exhibiting an attention pattern concentrating on the first_tok.
> > >
> > > Figure A4 ---- Second, position-based modular subtraction is in line with the observation that first_tok heads are prevelant beyond the first layer. Note that first\_tok heads do not unanimously attend to the first\_tok. Instead, they are selective for position indices with specific mod-10 values. Encoding such **fourier features** allows multiple first\_tok heads to collaboratively attend to the first token and may help subsequent layers in performing modular arithmetic.

---

> > > > ### Comment · Reviewer_oPHk · 2024-11-28
> > > > **Thanks for your response**
> > > >
> > > > My concerns have been addressed. Hope you will incorporate this evidence in the next version of your paper. I will raise my score.

---

> > > > > ### Author Response · Authors · 2024-11-28
> > > > > **We’re glad to hear that your concerns are addressed.**
> > > > >
> > > > > Thanks again for your help in strengthening our work!

---

> ### Author Response · Authors · 2024-11-20
>
> **3. Lack of novel concepts or methodologies.**
>
> Our work does not aim to introduce new methodologies and very explicitly avoids the introduction of new concepts as our work is evidence-based with a focus on making explicit the mismatch between claims in the literature and the lack of rigorous evidence and analysis. **Our novel contributions lie in empirical evidence and analysis.**
>
> We respectfully point out that our core contributions may not be fully recognized in your review and we’d like to discuss it further. To make discussion easier, we organize them into bullet points below, especially emphasizing those that are already recognized by other reviewers.
>
> - This work operationalizes the assessment of counting ability in LMs with well-designed tasks and well thought through experiments (jHk7, Funv)
> - Comparing counting performance across positional embeddings is a new contribution. (jHk7, Funv)
> - We not only provide the results, but dig deeper into the hidden factors that help explain the results, e.g. representing implicit BoS (RT8E), position-based modular arithmetic (jHk7),
> - Performing analysis from the perspective of what inductive biases are encoded by PEs or the natural recurrence in RNNs, which consequently facilitate counting, is interesting and novel (jHk7, Funv).
> - Overall, our work offers two high-level messages: 1) counting is not a built-in skill in most LM architectures, 2) counting could be imbued via certain architectural components that encode the relevant inductive bias. The first message calls for the field to reexamine what types of computation are assumed to be atomic. The second message suggests a potential direction for ***acquiring*** basic counting skills now that they cannot be taken for granted.
>
> **4. Insufficient contribution compared to what’s already studied by “When can Transformer count to n”**
>
> Despite similarity suggested by the title, the referenced paper “When can Transformer count to n” differs from our work in multiple key aspects (detailed below). So we believe that our work and theirs nicely complement each other and that both are important for guiding and shaping future work.
>
> - **Different axes of empirical comparison.** They did not focus on the PE dimension for comparison (they mainly used APE and NoPE). Nor did they compare against RNN-based architectures.
> - **Different task formats.** Their QC task with the “histogram solution” is similar to our selective counting task, but has a fixed input sequence length, thereby not considering extrapolation. The “count attend solution” is theoretically proved to be able to handle variable length, but there are no accompanying experiments, nor is it clear whether the ability to handle variable length can extrapolate to ***unseen*** lengths. Their MFE task only concerns counting the most frequent element rather than keeping counts for every element.
> - **Different analysis focus.** Their analysis focuses on the intricate relationship between hidden dimension and number of unique elements to count, while our analysis is centered around inductive biases needed for counting inductively.

---

### Official Review · Reviewer_jHk7 · 2024-11-03

**Soundness:** 4
**Presentation:** 4
**Contribution:** 4
**Rating:** 8
**Confidence:** 4

**Summary:**

This paper studies the ability of language models to learn counting inductively. Specifically, whether they can generalize counting beyond their training data. The experiments train Transformers with various positional embedding strategies (APE, SinePE, RoPE, SPE, NoPE). The core finding is that Transformers require specific positional embedding configurations to count effectively, while traditional RNNs handle counting tasks more naturally. The experiments reveal that different positional embedding approaches have distinct strengths: RoPE excels at unbounded counting with shifted starts, SinePE and APE perform well on both modular and selective counting, while NoPE and SPE are effective only for selective counting. These findings demonstrate that different architectural choices and embedding strategies significantly impact a model's ability to learn counting inductively, with implications for how counting capabilities should be implemented in language models.

Generally, this paper was very well written and easy to read. I would say that the biggest weakness is in the introduction and framing of the paper, where the formatting is unclear and the strong experimental results don't come through. If you can improve the first couple pages, this should be an excellent paper.

**Strengths:**

The generalization splits and counting tasks seem reasonable to me, clearly thought out well. The different performance across all of the kinds of positional encodings is very interesting, and I love the contrast between modular counting and typical unbounded state counting.

It's really interesting to focus on the positional embeddings as the design choice that might build in an inductive bias towards being able to count. It seems like core cognitive skills are implicitly built into different architectures, and perhaps papers like this one will lead to something exciting and new!

**Weaknesses:**

The abstract is very wordy and spends too much time explaining why counting is important. I would shorten the first half of the abstract to two or three sentences.

The quote at the beginning of the introduction, I’m not sure who the quote is being attributed to.

I am surprised you didn’t cite the bootstrapping counting paper by Steve Piantadosi, Josh Tenebaum, and Noah Goodman. This seems like an important citation, as this paper makes good on Carey’s earlier ideas in a computational framework.

The formatting for the inductive counting principle makes it unclear where this principle is coming from. Is it from an earlier work, or is it just a general idea that you are making precise here? The reference to ordered lists of words makes the definition seem a bit idiosyncratic, but maybe I’m wrong about this!

**Questions:**

I would really love to see you take this analysis into the interpretability setting. With controlled datasets like this, tools like activation patching a distributed alignment search could be used to localize counting states within network representations, and then you could get a mechanistic analysis of how this ability develops!

---

> ### Author Response · Authors · 2024-11-20
> **Thank you so much for your thoughtful questions and positive feedback!**
>
> **1. Framing of the first couple pages**
>
> Thank you for your deep engagement with the work.  We realize this is a rather intricate paper to write and so we’d be happy to improve and polish any of the writing and would love specific recommendations or points which you feel are unclear.  Any framing recommendations will further strengthen the work!
>
> **2. Shorten first half of the abstract**
>
> Thanks for the suggestion. We have revised the abstract (see the updated submission)
>
> **3. Missing reference to the bootstrapping counting paper.**
>
> The bootstrapping counting paper is indeed very relevant to our work. We have now cited it in our updated manuscript.
>
> **4. Elaboration on the principles that lead to our formatting of counting, and the source of the quote.**
>
> The principles that lead to our formatting of counting borrow heavily from the cogsci literature studying how children learn the numerical meaning of number words.
>
> It is proposed that children initially develop mental representations for “sets of cardinality k”, distinctly from how they represent number words (as they hear utterances of “one”, “two”, “three” …). At the subset-knower level, a learner can 1) routinely utter a list of number words up to N (N > 3); 2) associate their mental models of cardinality to **only the first few number words**, e.g. “one”, “two”, “three”. At the cardinal-principle(CP)-knower level, a learner discovers the abstract relationship between the list of number words and their meanings for cardinality. This suggests that knowing the list of number words up to N is a necessary but not sufficient condition for the ability to count up to N. The missing part is some inductive mechanism that captures the conceptual change from modeling finite associations to generalization towards infinity.
>
> Transferring the insights to the LM context, “succession”, “helper token” and “shiftedstart” formats suggest three possible ways to teach the model the ordering of number words, thus satisfy the necessary condition. Yet the poor model performance verifies that knowing the number word list isn’t sufficient for knowing how to count.  “helper token” is reminiscent of a scenario where a human learner receives word-cardinality supervision more frequently with certain objects than with other objects. Yet the learner has to generalize such word-cardinality mapping to all countable objects. “shiftedstart” is a modification tailored to the consideration that a model is implemented with a finite context window.
>
> Source of the quote: Rips, Lance J. ; Asmuth, Jennifer & Bloomfield, Amber (2006). Giving the boot to the bootstrap: How not to learn the natural numbers. Cognition 101 (3):B51-B60. We cite this paper in our updated manuscript.
>
> The original quote reads:  `If "k" is a number word that refers to the property of collections containing n objects, then the next number word in the counting sequence "next (k)" refers to the property of collections containing one more than n objects.`
>
> It is also quoted in: Piantadosi, Steven T, Joshua B. Tenenbaum and Noah D. Goodman. “Bootstrapping in a language of thought: A formal model of numerical concept learning.” *Cognition* 123 (2012): 199-217.
>
> **5. Use activation patching to localize and trace the development of counting states.**
>
> Thanks so much for this suggestion. We agree there is an intimate connection between our work and the mechanistic interpretability literature. We have attempted to interpret our Transformer models via attention-visualization and clustering on intermediate states. We did not use more sophisticated techniques like activation patching or DAS because our models are shallow and each layer likely implements a distinct subtask, rather than progressively promoting the confidence of the final counting result. This has been indicated by our preliminary analysis using logit lens. The logit lens shows that the counting state only exists in the 2nd layer of a 2L Transformer, or the 3rd-4th layer of a 4L Transformer. In the future, it is definitely worth investigating deeper models handling more complex tasks that subsume counting. In such a scenario, mechanistic interpretability tools can be very helpful in tracing the implicit development of counting states.

---

### Meta-Review · Area_Chair_fikk · 2024-12-17

**Metareview:**

This paper investigates the claim that Transformers can count. Through careful experimentation, it shows that they can’t, like modern RNN architectures, while traditional RNN architectures can. This paper shows a surprising result about a fundamental capacity of sequence models supported by careful experiments.

The reviewers appreciate the significance of the topic and the rigorousness of the experiments and vote for acceptance. Reviewer oPHk’s concerns about the lack of insights have been addressed by visualizations and as a result they increased their score. Reviewer RT8E is concerned about the ability to draw insights from training toy transformers on toy data to practical settings. This concern has been partially addressed.

**Additional Comments On Reviewer Discussion:**

-

---

### Decision · Program_Chairs · 2025-01-22

Accept (Poster)